# Occlusive membranes for guided regeneration of inflamed tissue defects

Woojin Choi [1,8], Utkarsh Mangal [2,8], Jin-Young Park[3], Ji-Yeong Kim [2], Taesuk Jun [1], Ju Won Jung[4], Moonhyun Choi [1], Sungwon Jung[1], Milae Lee [1], Ji-Yeong Na[3], Du Yeol Ryu [1], Jin Man Kim[4], Jae-Sung Kwon [5], Won-Gun Koh[1], Sangmin Lee[6], Patrick T. J. Hwang [7], Kee-Joon Lee[2], Ui-Won Jung[3], Jae-Kook Cha [3] ✉, Sung-Hwan Choi [2] ✉ & Jinkee Hong [1] ✉

Guided bone regeneration aided by the application of occlusive membranes is a promising therapy for diverse inflammatory periodontal diseases. Symbiosis, homeostasis between the host microbiome and cells, occurs in the oral environment under normal, but not pathologic, conditions. Here, we develop a symbiotically integrating occlusive membrane by mimicking the tooth enamel growth or multiple nucleation biomineralization processes. We perform human saliva and in vivo canine experiments to confirm that the symbiotically integrating occlusive membrane induces a symbiotic healing environment. Moreover, we show that the membrane exhibits tractability and enzymatic stability, maintaining the healing space during the entire guided bone regeneration therapy period. We apply the symbiotically integrating occlusive membrane to treat inflammatory-challenged cases in vivo, namely, the open and closed healing of canine premolars with severe periodontitis. We find that the membrane promotes symbiosis, prevents negative inflammatory responses, and improves cellular integration. Finally, we show that guided bone regeneration therapy with the symbiotically integrating occlusive membrane achieves fast healing of gingival soft tissue and alveolar bone.

Tooth loss can occur accidentally or by infectious periodontal diseases, primarily periodontitis. It leads to alveolar atrophy, translating into the loss of periodontal soft tissue (gingival gum) and hard tissue (alveolar bone)[1]. Because an adequate alveolar bone volume is a prerequisite for successful implant treatment, guided bone regeneration (GBR) therapy has been performed on diverse edentulism patients[2]. For instance, repairing critical size defects using GBR technology has led to the success of dental implant procedures, with over 40% incidence of clinical use[3,4]. Therefore, effective and long-term results of GBR is necessary for the broad dentistry fields. Recently, the predictable outcome of GBR has been achieved through advanced occlusive membrane technologies[5]. Occlusive membranes inhibit fibroblasts and epithelial cells from invading the periodontal alveolar defect and preferentially retain the space for osteoblast growth[1,6].

[1]Department of Chemical and Biomolecular Engineering, College of Engineering, Yonsei University, Seoul 03722, Republic of Korea. [2]Department of Orthodontics, Institute of Craniofacial Deformity, Yonsei University College of Dentistry, Seoul 03722, Republic of Korea. [3]Department of Periodontology, Research Institute for Periodontal Regeneration, Yonsei University College of Dentistry, Seoul 03722, Republic of Korea. [4]Department of Oral Microbiology and Immunology, School of Dentistry and Dental Research Institute, Seoul National University, Seoul 08826, Republic of Korea. [5]Department and Research Institute of Dental Biomaterials and Bioengineering, BK21 FOUR Project, Yonsei University College of Dentistry, Seoul 03722, Republic of Korea. [6]School of Mechanical Engineering, Chung-ang University, 84, Heukserok-ro, Dongjak-gu, Seoul 06974, Republic of Korea. [7]Cardiovascular Institute, Rowan-Virtua School of Translational Biomedical Engineering & Sciences, Rowan University, 201 Mullica Hill Rd., Glassboro, NJ 08028, USA. [8]These authors contributed equally: Woojin Choi, Utkarsh Mangal. ✉e-mail: chajaekook@yuhs.ac; selfexam@yuhs.ac; jinkee.hong@yonsei.ac.kr

During GBR procedures, the cell-rich hard tissue region—the inner zone covered with the occlusive membrane—undergoes closed healing. Conversely, the gingival soft tissue area—the outer side of the occlusive membrane—is microbially exposed and undergoes open healing (Fig. 1a). Considering these complex regeneration scenarios, a practical occlusive membrane should possess sufficient mechanical properties, be clinically tractable, be able to integrate with the host's tissues, and be able to create healing space[6,7].

Current occlusive membranes are either non-resorbable or resorbable and have distinct limitations[6]. The non-resorbable membranes, e.g., polytetrafluoroethylene membranes and titanium meshes, are excellent space maintainers. However, they present low biological permeability and can induce inflammatory responses in the body. Even worse, additional surgery is required to remove the deployed membrane. Hence, resorbable membranes, e.g., collagen membranes (CMs), are clinically preferred. Although the resorbable membranes integrate with the host's tissues, which eliminates the need for removal surgery, they tend to have insufficient dimensional and structural stability[8]. Recent studies have incorporated bioactive reagents, such as pharmaceutical agents, hormones, or cations, to impart cellular affinity, broad-spectrum antimicrobial properties, or longevity to these membranes[9–12].

In natural, symbiosis, a healthy homeostasis between the host microbiome and cells, occurs in the oral environment (Fig. 1b)[13]. In other words, both microbes and cells mutually thrive in the periodontal organization. To date, occlusive membrane technologies have primarily considered the competitive inhibition between fibroblasts

and osteoblasts[1]. The homeostatic oral microbiome endogenously involves the pathobionts, which could promote disease under specific conditions[14]. In this context, after being affected by the oral microbiome, occlusive membranes can cause adverse complications (i.e., secondary inflammatory diseases) during the GBR process (Fig. 1c)[15]. Thus, the third subject, the oral microbiome, should be contemplated. During GBR periods of several months, the symbiotic biology of the periodontal alveolar region is prone to be disturbed, which could delay predictable and timely healing[15]. Generally, the outer open healing zone experiences microbial intrusion and a selective increase in healing-suppressing pathobionts[16]. After this premature loss, the occlusive membrane becomes the next microbial culture bed. The removal, loss, or premature resorption of the microbiome-affected occlusive membrane can transmit the infection into the inner closed healing zone[17]. Considering this specific periodontal structure, an advanced occlusive membrane should be able to induce healthy symbiosis during the entire GBR period. In conclusion, the symbiotic integration of occlusive membranes, i.e., long-term integration while maintaining a periodontal symbiosis, is necessary for achieving next-generation GBR therapy (Fig. 1d).

Our tooth endogenously retains symbiosis regardless of the diverse microbes in the oral cavity. In particular, the tooth's outer layer, i.e., enamel, significantly contributes to the maintenance of a healthy oral microbiome. The pathobionts, representatively, *Streptococcus* and *Porphyromonas*, also dwell in the homeostatic biofilms on exposed tooth surfaces[13]. If these keystone pathogens form harmful colonization, they demineralize the tooth enamel, construct the

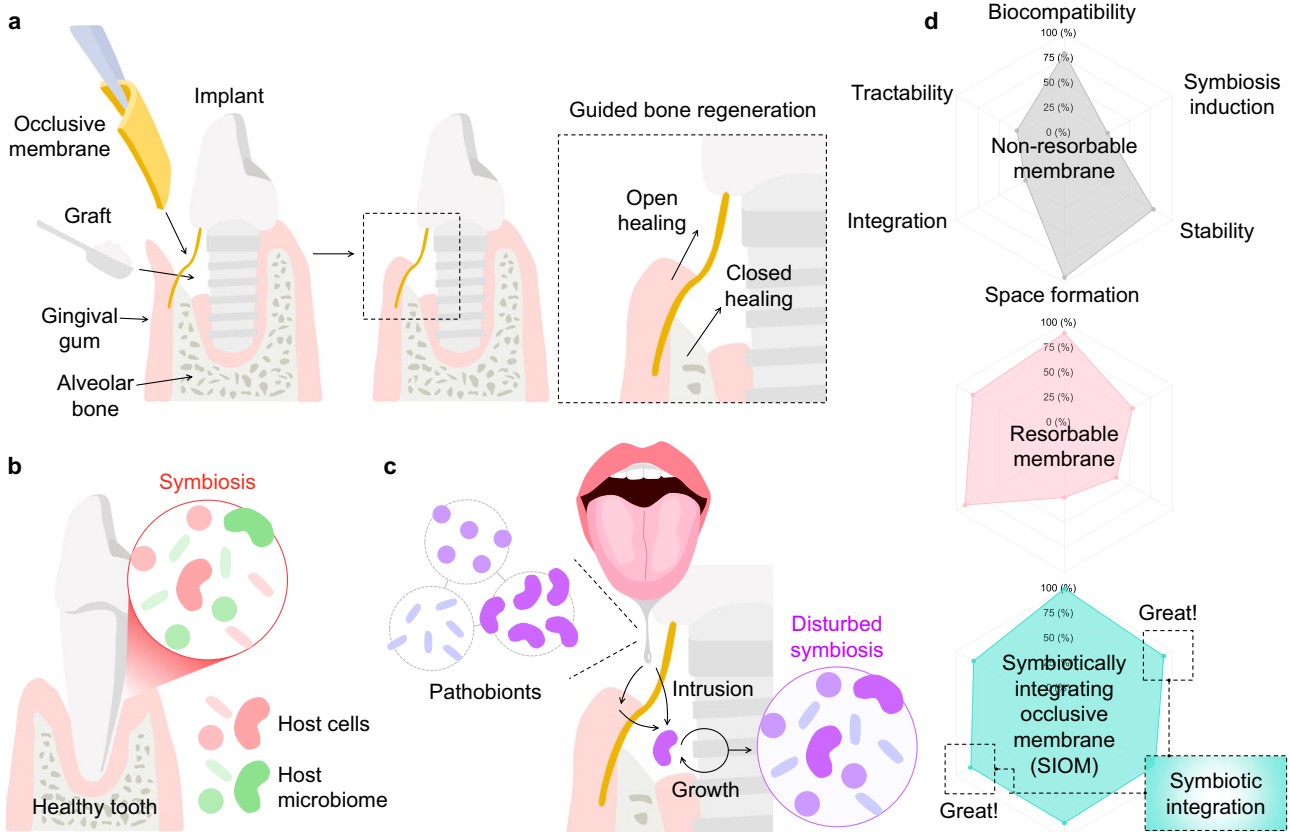

**Fig. 1 | Significance of the SIOM in GBR therapy. a** Schemes of the general GBR process for the periodontal tissue with a tooth implant. The occlusive membrane was deployed as a boundary between the open healing of the soft tissue (gingival gum) and the closed healing of the hard tissue (alveolar bone). **b** Symbiosis status of the healthy periodontal organization. Host cells and microbiomes symbiotically exist through cooperative interactions. **c** Scenario

depicting the disturbance of the periodontal symbiosis. Microbial intrusion results in microbial–cellular competition and secondary complications. **d** Radar charts of the non-resorbable membrane, resorbable membrane, and SIOM. The essential attributes of an ideal occlusive membrane are biocompatibility, symbiosis-inducing effects, biological durability, space formation, integration with periodontal tissues, and tractability.

anaerobic cavity, aggregate into acidic caries, and finally damage the entire oral microbiome[15,18]. While pathogenic colonization is controlled by several factors (*e.g.*, sugar-rich diet, salivary condition, oral hygiene)[19], the tooth enamel physically prevents demineralization and cavitation through its robust barrier structure: durable and dense hydroxyapatites (HAPs).

Inspired by this naturally maintained symbiosis, we developed an occlusive membrane that is entirely analogous to tooth enamel, from the HAP growth mechanism to dense barrier HAPs. To this end, we established enamel-mimetic multiple nucleation biomineralization of HAP in a resorbable CM. The obtained membrane is called a symbiotically integrating occlusive membrane (SIOM). When the SIOM was employed for the GBR therapy, it might construct a considerably enamel-like environment at the interface of closed healing and open healing regions, inducing symbiosis. To prove this concept, the SIOM was subjected to human saliva and in vivo canine experiments to gather bulk microbiome data. We discovered that the SIOM induced the symbiosis by causing an early normobiosis effect on the post-treatment microbiome community. In particular, the SIOM was administrated to regenerate the soft and hard tissue in the inflammatory-challenged microenvironment after the diseased canine premolars

were extracted in vivo. Remarkably, the SIOM induced symbiotic conditions during the open and closed healing phases of GBR therapy. Finally, it integrated into the challenging oral environment and promoted soft and hard tissue regeneration, yielding the ideal GBR therapy prognosis.

## Results

### Multiple nucleation biomineralization to prepare the SIOM entirely composed of HAP

We used the natural biomineralization strategy for tooth enamel to prepare the SIOM, which fully composed of HAP[20]. Tooth enamel is comprised of dense HAP biomineralized via controlled ion adsorption (Fig. 2a)[21,22]. In particular, the osteopontin (OPN) favorably interacts with $PO_4^{3-}$ ion, while dentin matrix acidic phosphoprotein 1 (DMP1) captures $Ca^{2+}$ [23,24]. These selective ion adsorptions enable a characteristic enamel growth, *i.e.*, multiple nucleation biomineralization for dense coverage of HAP. Accordingly, we developed a selective ion-adsorbing zwitterionic layer in the CM. (Fig. 2b). This zwitterionic membrane (ZM) was fabricated by conducting dual radical-based zwitterionic sulfobetaine polymerization on the CM (Supplementary Fig. 1a)[25]. The significant zwitteration of the CM was confirmed using

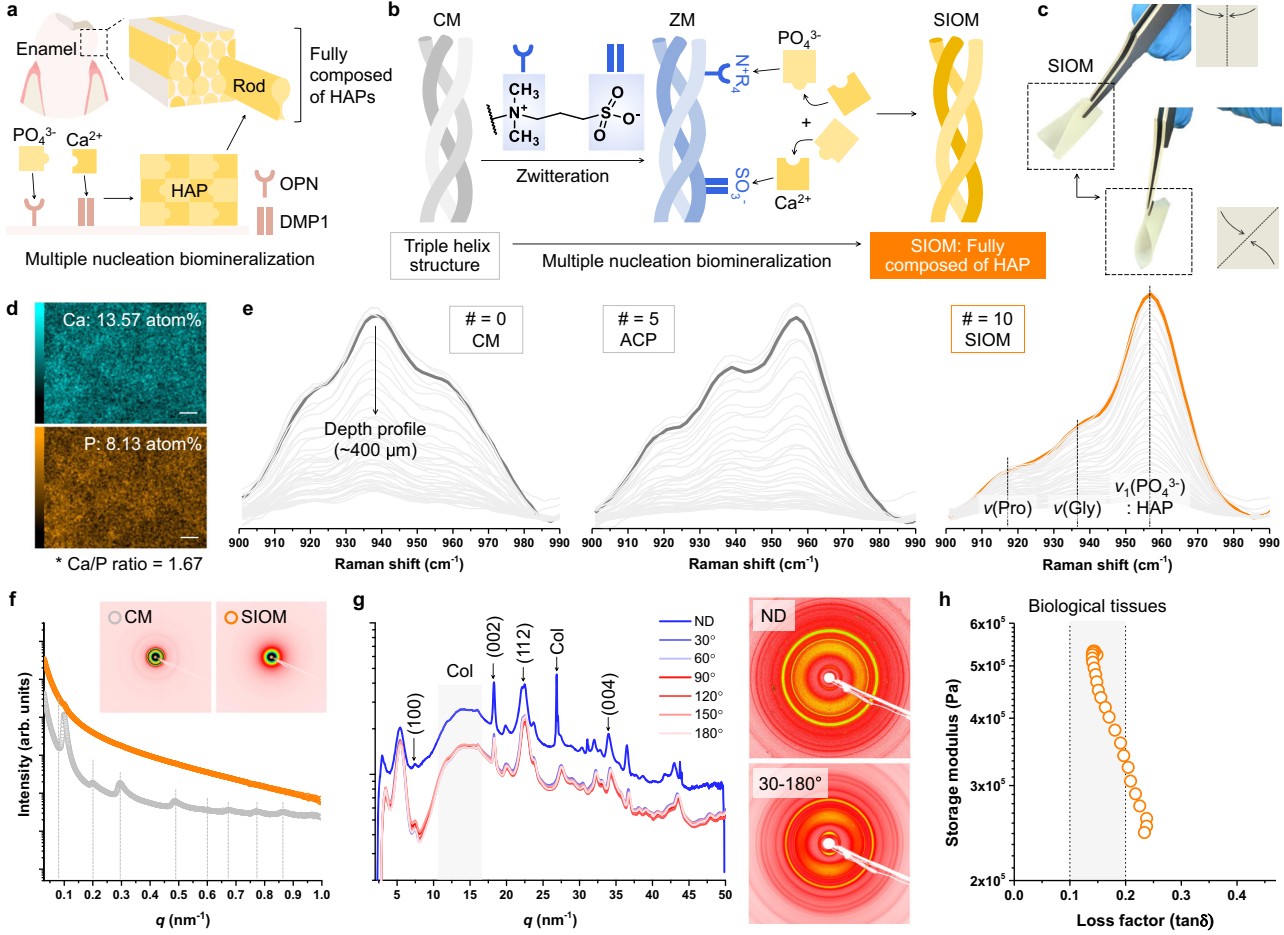

**Fig. 2 | Comprehensive investigation of the SIOM. a** Natural multiple nucleation biomineralization process for constructing dense tooth enamel structures. **b** Schemes depicting the osteo-mimetic zwitteration of the CM into a zwitterionic membrane (ZM) and the multiple nucleation biomineralization process emulated to obtain an entirely biomineralized SIOM. **c** Representative photographs of the SIOM (1 × 1 cm²). **d** Ca and P ion mapping of the SIOM acquired using an energy-dispersive spectrometer. Scale bar = 2 μm. **e** Raman spectra at 900–990 cm⁻¹. The light gray lines indicate the depth profiles measured per 10 μm up to 400 μm. The black and orange bold lines indicate

the top surface results. ACP represents amorphous calcium phosphate. **f** One-dimensional and two-dimensional small-angle X-ray scattering (SAXS) patterns of the CM and SIOM. *q* is the scattering vector, where $q = (4\pi/\lambda) \sin(\theta/2)$. *θ* and *λ* are the scattering angle and wavelength of the incident X-ray beam, respectively. **g** One-dimensional and two-dimensional wide-angle X-ray scattering (WAXS) patterns of the SIOM. X-rays were irradiated in the normal direction (ND; *z* axis) and cross-sectional direction (*x*–*y* axis) of the SIOM. **h** Storage modulus *vs.* loss factor (tanδ) plot of SIOM. Source data are provided as a Source Data file.

X-ray photoelectron spectroscopy, ion mapping, and time-of-flight secondary ion mass spectrometry (Supplementary Fig. 1b–e). The zwitterionic groups functioned as nucleation sites, similar to OPN and DMP1. Specifically, the positive moiety of the ZM ($-N^+R_4-$) preferentially interacted with the $PO_4^{3-}$ ions, and the negative moiety ($-SO_3^-$) was bound to $Ca^{2+}$. These selective ion adsorptions determined the biomineralization kinetics, as shown in Supplementary Fig. 2. Thus, the ZM enabled the multiple nucleation biomineralization and full coverage of HAP within the SIOM, as will be discussed in the following sections[26].

The ZM was repeatedly exposed to 0.3 M $PO_4^-$ and 0.5 M $Ca^{2+}$ to advance its biomineralization into the SIOM[27]. Fig. 2c shows the image of the prepared $1 \times 1\,cm^2$ SIOM. The SIOM was tractable and could be readily used for GBR (Supplementary Fig. 3). The SIOM had a Ca/P ratio of 1.67, indicating that thermodynamically stable HAP [$Ca_{10}(PO_4)_6(OH)_2$] was generated (Fig. 2d). In particular, the SIOM was obtained after undergoing ion exposure 10 times (# = 10, where # is the number of $PO_4^{3-}-Ca^{2+}$ exposure sets). The entire depth Raman profiles were monitored by increasing # from 0 (CM) to 10 (SIOM) (Fig. 2e). The CM exhibited apparent Raman signals of proline and glycine at ~920 $cm^{-1}$ and ~935 $cm^{-1}$, respectively. The HAP signal at ~955 $cm^{-1}$ intensified when # was 5, implying that amorphous calcium phosphate was generated[26]. When # reached 10, the HAP signal of the SIOM intensified significantly. The X-ray diffraction spectrum of the SIOM displays the characteristic HAP pattern (Supplementary Fig. 4).

Subsequently, the CM and SIOM were investigated using SAXS (Fig. 2f). The SAXS pattern revealed that the CM had a lamellar structure with a d-spacing of 63 nm[28]. However, this lamellar structure disappeared after the CM biomineralized into a SIOM. WAXS experiments were conducted, as shown in Fig. 2g. X-rays were transmitted to the SIOM in the normal direction. Thereafter, cross-sectional WAXS signals were obtained by rotating the X-ray by 30° at the $x–y$ plane of the SIOM. Regardless of the X-ray incidence angle, the SIOM exhibited distinct HAP peaks, namely, (002), at 18.3 $nm^{-1}$. The mercury intrusion porosimeter experiments suggested that the SIOM featured a denser submicron pore structure than CM (Supplementary Fig. 5). Accordingly, we concluded that the entire CM was completely biomineralized into a SIOM. Moreover, the fully covered HAP compensated for the free radical-induced decreased mechanical property of the ZM (Supplementary Fig. 6). In other words, the application of SIOM means an establishment of a durable and tooth enamel-like ordered structural environment during GBR therapy.

When a developed material is applied to a specific tissue, it should exhibit similar viscoelasticity to surrounding tissues for a satisfactory operation[29]. The viscoelasticity of SIOM was studied through the storage modulus *vs.* loss factor profile (Fig. 2h)[30,31]. The loss factors of most biological materials involving extracellular matrix, soft tissues, and hard tissues are in the range of 0.1–0.2[32]. Remarkably, the viscoelasticity of the SIOM was comparable to the biological tissues. Therefore, SIOM is physically beneficial when deployed at the interface of the gingival gum and the alveolar bone defect.

## Structural stability of the SIOM under enzymatic degradation conditions

During several months of GBR therapy, the occlusive membrane should maintain its structure to provide sufficient time and space for alveolar bone regeneration. In the early phase of regenerative healing, catabolic enzymes and primary response inflammatory cytokines are generated, which affects the stability of the occlusive membrane[33,34]. Although collagenase type I is a necessary regeneration-associated enzyme, it causes the rapid degradation of CMs, collapsing the closed healing zone. Accordingly, the enhanced enzymatic stability of SIOM should be established. Moreover, we functionalized the CM with various charge groups (*R*) to evaluate the significance of osteo-mimetic multiple nucleation biomineralization in yielding outstanding stability

(Fig. 3a). Negative membrane (NM) and positive membrane (PM) were prepared with two strong polyelectrolytes: polystyrene sulfonate and poly(diallydimethylammonium chloride). ZM-1 is a sulfobetaine zwitterionic polymer, the original component of the SIOM. ZM-2 is a phosphorylcholine zwitterionic polymer. Subsequently, the NM, PM, ZM-1, and ZM-2 were biomineralized in the same manner as the SIOM, *i.e.*, # = 10 with 0.3 M $PO_4^-$ and 0.5 M $Ca^{2+}$. The NM and PM possessed a single nucleation site. Therefore, HAPs were grown into spherically aggregated structures via single nucleation biomineralization (Fig. 3b)[35]. Owing to the sparsely distributed granular HAPs, the collagen substrate was still visible on the biomineralized NM and PM surfaces. In contrast, ZM-1 and ZM-2 underwent multiple nucleation biomineralization, producing fully covered wrinkled flake HAPs. In summary, osteo-mimetic multiple nucleation biomineralization is essential for the full coverage of HAPs within SIOM.

The densely grown HAPs physically and electrostatically blocked collagenase penetration, thereby improving the enzymatic stability of the SIOM (Fig. 3c). To prove this concept, the pristine CM and the biomineralized NM, PM, ZM-1, and ZM-2 were immersed in collagenase type I media and incubated at 37 °C (Fig. 3d)[36]. The CM degraded rapidly within 12 h, demonstrating its practical limitation with spacing durability. Meanwhile, the CM exhibited excellent stability for hydrolytic degradation (Supplementary Fig. 7). As shown in Fig. 3e, the degradation rate ($h^{-1}$) was determined by assessing the residual membrane within 12 h. The biomineralized NM and PM presented slightly decreased degradation rates: 0.035 $h^{-1}$ of CM, 0.025 $h^{-1}$ of NM, and 0.029 $h^{-1}$ of PM. However, they degraded after 12 h of enzyme exposure since the sparsely distributed HAPs could partially block collagenase penetration. Meanwhile, ZM-1 and ZM-2 with fully covered wrinkled flake HAPs did not degrade within 24 h, resulting in a degradation rate <0.005 $h^{-1}$. Interestingly, the residual biomineralized ZM-1 (*i.e.*, SIOM) was monitored even after 48 h of degradation. When the CM and SIOM were immersed in an α-amylase solution, a remarkable improvement in their enzymatic stability was also observed (Supplementary Fig. 8). Moreover, the storage and loss moduli of the CM and SIOM were measured during 10 h of degradation (Fig. 3f). The CM lost ~90% of its storage modulus after 4 h of degradation, implying that the GBR zone dented rapidly. On the other hand, the SIOM maintained its original properties (e.g., shear modulus and Ca/P ratio of 1.67) even after 10 h of degradation. Accordingly, the SIOM exhibited outstanding enzymatic stability and long-lasting space formation properties.

## Symbiosis-inducing effect of the SIOM

The SIOM was subjected to Good Laboratory Practice-ensured cytocompatibility tests (ISO 10993-5: 2009). Slight grade 1 reactivity was observed within a non-discernible zone, confirming the highly biocompatible performance of the SIOM (Supplementary Table 1–2). Progenitor cells, namely, BMSCs, were used to evaluate the osteoconductivity of the SIOM to confirm its cellular symbiosis-inducing performance. BMSCs and CM or SIOM were cultured in growth media and osteogenic differentiation media for 2 weeks. Subsequently, a quantitative real-time polymerase chain reaction was performed to study the expression degree of osteogenic biomarkers. Interestingly, OPN expression increased 2.67-fold in BMSCs cultured in the growth media with the SIOM, thereby indicating the increased density of osteogenic differentiation (Fig. 4a). According to BMSC tests conducted under differentiation media conditions, the SIOM exhibited significantly enhanced expression of differentiation markers: a 127% increase in RUNX2, a 118% increase in OCN, a 1065% increase in OPN, and a 2884% increase in BMP2 (Fig. 4b). The above biomarker results indicate that the osteo-mimetic SIOM mediated an excellent stimulation for osteo-differentiation[37]. In particular, the significantly enhanced expressions of OPN and BMP2 implied that SIOM promotes bone healing up to the bone tissue formation stage during 2 weeks.

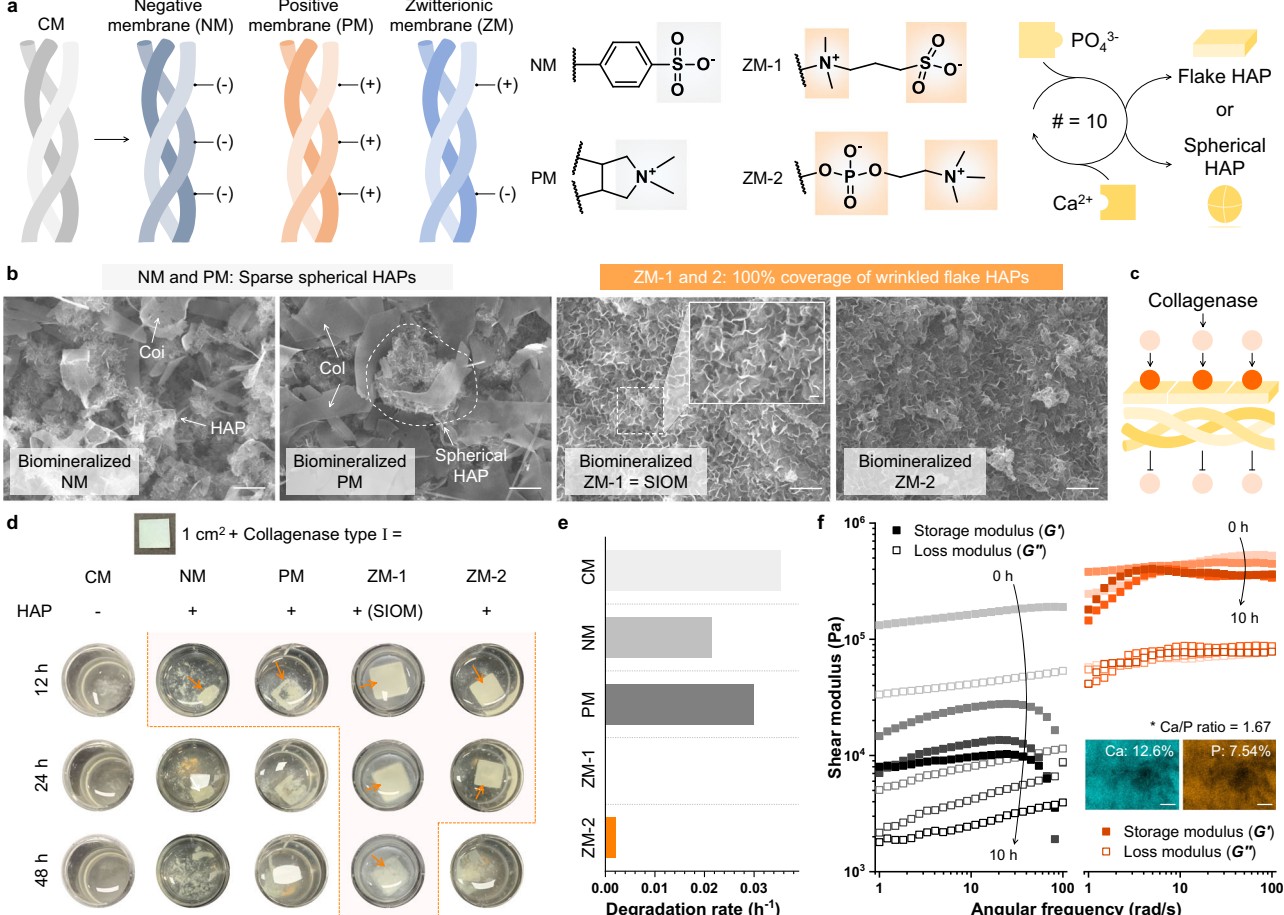

**Fig. 3 | Evaluation of enzymatic durability. a** Functionalization of the CM with different charge groups. The chemical structures of a negative membrane (NM), a positive membrane (PM), and two ZMs are suggested. The characteristics of the substrate determine the HAP geometry. **b** Scanning electron microscope images of the biomineralized NM, PM, and ZM-1. Scale bar = 1 μm. *n* = 5 images per membrane group. **c** Schemes of collagenase penetration through the biomineralized NM, PM, and ZM. **d** Images of the CM and the biomineralized NM, PM, ZM-1, and ZM-2 after being immersed in collagenase media (0.2 IU mL⁻¹). The orange arrows indicate the residual membranes. **e** Degradation rates of the CM and the biomineralized NM, PM, ZM-1, and ZM-2. **f** Rheological responses of the CM (left) and SIOM (right) during 10 h of enzymatic degradation. The inserted ion mapping image was obtained from the SIOM after the end-point of collagenase exposure. Scale bar = 2 μm. Source data are provided as a Source Data file.

Therefore, SIOM is desirable for the GBR therapy of several months requiring the expeditious augmentation of the alveolar bone.

The nature of the microbiota-contacting substrate closely determines the microbial ecology in the vicinity[38]. Here, the symbiosis-inducing effect of the SIOM was investigated by monitoring whether a healthy microbiome community was constructed at the SIOM. To study the microbiome development at the SIOM, we used 16 S rRNA sequencing of co-cultured salivary extracts from six healthy adults (Fig. 4c). The observed operational taxonomic units in the sequencing showed a distinct early plateauing curve for the ZM and SIOM with low sequencing, indicating a genuine lack of diversity (Supplementary Fig. 9)[39]. As shown in Fig. 4d, the *Firmicutes*-to-*Bacteroidetes* (F/B) ratio increased significantly (*ca.* 49.0-fold) during the microbial community level interaction at the ZM and SIOM. Specifically, the gram-negative classes of *Bacteroidetes* reduced remarkably (Supplementary Fig. 10). This F/B profile is associated with competition for the metabolic fermentation pathway and interacts negatively in the human gut[40]. Moreover, a low F/B gut profile has been co-associated with increased pro-inflammatory cytokine expression[41,42]. Thus, a settled microbiome community with an unilaterally low F/B scan may impede normal healing through a chronic expression of pro-inflammatory cytokines.

As shown in the relative abundance at genus level (Supplementary Fig. 11), three periodontal pathobionts serving as taxonomic biomarkers, namely, *Prevotella*, *Allopreveotella*, and *Actinomyces*, did not exhibit any abundance at SIOM[43,44]. More specific observations could also be made on resolving the differences at the species level resolution (Fig. 4e). The species with pathobiont nature, *Fusobacterim periodonticum, Gemella haemolysans, Haemophilus parainfluenzae, Porphyromonas pasteri* and *Veillonella sps* groups, remained undetected from the SIOM (Supplementary Fig. 12). Furthermore, SIOM exhibited a lower abundance of the bridging species, *Veillonella*, which leads to dysbiosis by acting as an accessory pathogen for keystone pathobionts[45,46]. In particular, *Veillonella parvula, Veillonella dispar,* and *Veillonella rogosae* were found in higher concentrations in CM. This result is associated with the actual lack of gamma diversity as the Hill number increased (Supplementary Fig. 13)[39]. Typically, a shifted microbiome is described as a niche-specific species turnover (gain or loss). Accordingly, the near total lack of pathobionts (especially *Bacteroidetes* and *Actinobacteria*) indicates that the microbiome community at the SIOM prefers commensal oral microbionts[47].

Typically, the core microbiota is indicative of disease–disease relationship and it is varied by immediate environmental changes[48]. The SIOM had a high co-occurrence representation of genera with PC, proving its potential for recovering microbial communities (Fig. 4f). In contrast, the ZM formed unique taxa and core microbiota distribution: *Bacteroides vulgatus, Enterococcus faecium* groups, and *Romboutsia*

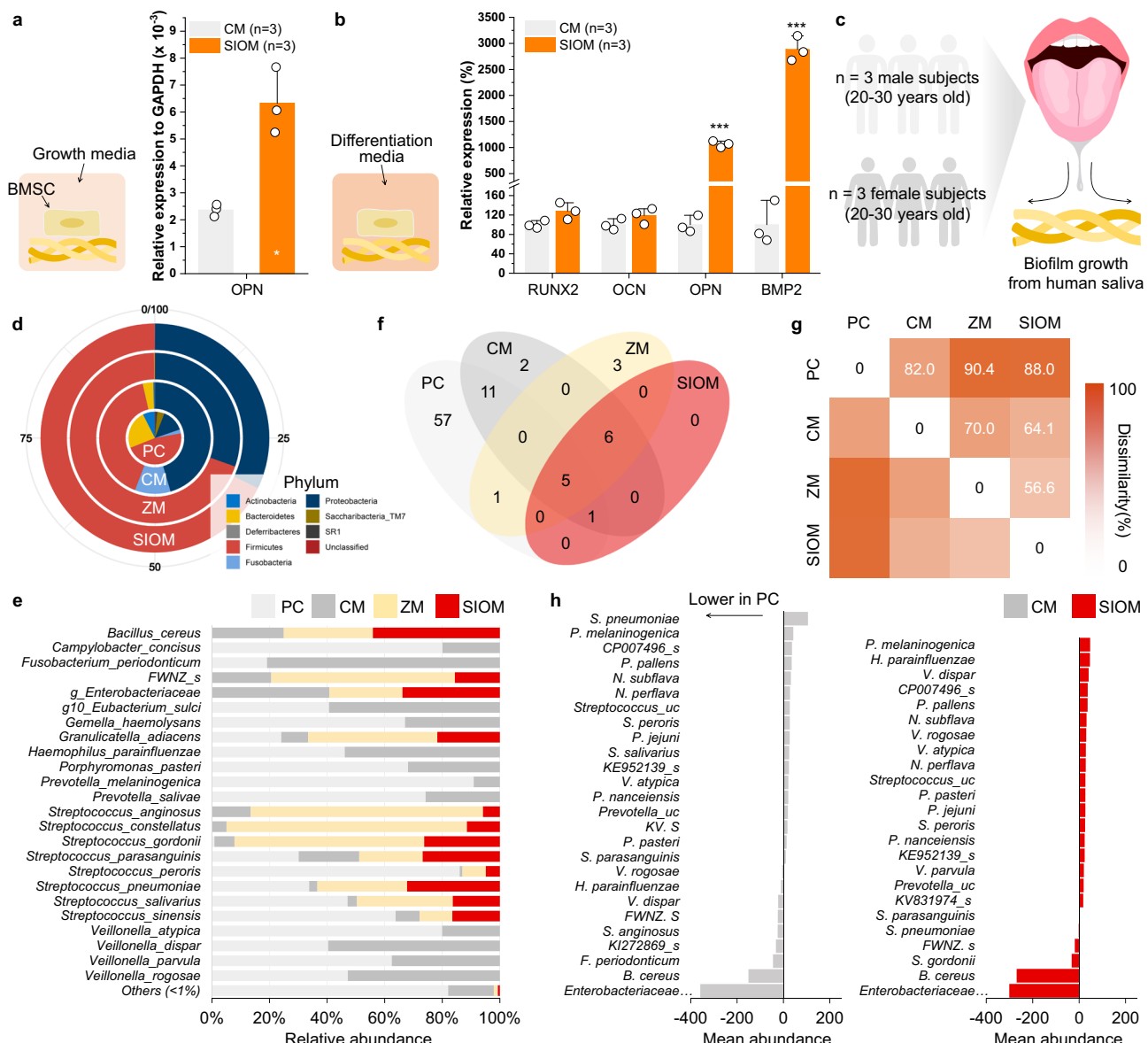

**Fig. 4 | Evaluation of symbiosis induction by the SIOM.** Analysis of biological responses from human bone marrow stem cells (BMSCs) via cytokine expression (RUNX2, OCN, OPN, and BMP2). BMSCs were cultured in the **a** growth media (mean ± SD, $n = 3$, two-tailed $t$-tests, $p = 0.0317$) and **b** osteogenic differentiation media (mean ± SD, $n = 3$, two-tailed $t$-tests, $p = 0.000012$ for OPN, $p = 0.000249$ for BMP2). **c** In vitro experiment protocol showing the collection of healthy human adult saliva for culture over the occlusive membranes for a 16 S RNA sequencing assay. Positive control (PC) means the group without an occlusive membrane. **d** Taxonomic composition for the CM, ZM, and SIOM at the phylum level in comparison to pooled saliva culture sequences. **e** Comparison of species level relative abundance for taxa exhibiting the highest prevalence. Taxa representing less than 1% of total sequences were grouped as others. **f** Core microbiota Venn plot presenting species level shared and unique taxa by overlapping regions. **g** The dissimilarity percentage between the groups against the pooled saliva reference computed using SIMPER analysis. **h** The differences in abundances of taxonomic groups with greater than 1% contribution to dissimilarity to PC. The negative values in $x$ axis indicate lower abundance in PC compared to CM/SIOM and vice-versa. RUNX2: runt-related transcription factor 2; OCN osteocalcin, OPN osteopontin, BMP2 bone morphogenic protein-2. Source data are provided as a Source Data file. $^*p < 0.05$, $^{**}p < 0.01$, $^{***}p < 0.001$.

*timonensis* (Supplementary Data 1). In this context, the ZM was insufficient to induce a healthy microbiome although its zwitterionic groups nonspecifically prevented microbial growth up to 50% (Supplementary Fig. 14)[49]. Hence, ZMs without osteo-mimetic multiple nucleation biomineralization may be the tipping point for disturbed microbiomes[50].

We also observed that the SIOM significantly impacted the overall diversity similarity index (Fig. 4g). The ZM and SIOM showed a high dissimilarity (>79%) to the PC reference. Most of the observed differences could be attributed to the variations in the core microbiota (>70% prevalence)[51]. Fig. 4h draws the comparison with the incubate of

native pooled salivary microbiota (PC), and Supplementary Fig. 15 shows the entire name of taxonomic groups. Regarding both CM and SIOM, the dissimilarity contribution was observed to be highest for *Enterobacteriaceae* and *Bacillus cereus* groups. However, *V.dispar, P. melaninogenica, and F. periodonticum* had ~3% dissimilarity contribution with higher abundance expressed at the CM interface. Furthermore, in specific periodontally related indicator species (Supplementary Note), the pathogenic *Porphyromonas* and *Fusobacterium* genera were observed in the PC and CM groups only.

However, due to the sampling of saliva from a healthy cohort, it was limited in identifying the keystone species for periodontal

pathogenesis using amplicon sequencing. Accordingly, we conducted *Porphyromonas gingivalis* and *Staphylococcus aureus* colony forming experiments to assess the specific microbial interaction (Supplementary Fig. 16). Remarkably, SIOM showed a 68% decreased *P. gingivalis* colony compared to the CM. Moreover, SIOM exhibited a 54% reduction in *S. aureus* colony formation. In particular, *S.aureus* is one of the pathogens with a favorable affinity towards titanium implants and found in the early microbiota after surgery[52]. Considering that SIOM deployed near dental implants, its strong resistance against *P. gingivalis* and *S. aureus* further confirms its enhanced performance for satisfactory GBR outcome. Supplementary Fig. 17 shows the unsupervised pattern analysis with the Spar CC method. Notably, the species positively correlated to the identified pathobionts (*P. pasteri, F. nuceatum,* and *F. periodonticum*) had low relative abundance in SIOM. In other words, the SIOM exhibited a weaker correlation with the pathobionts. In conclusion, we confirm that the SIOM had an improved symbiosis-inducing effect based on its favorable cellular and prokaryotic response.

## In vivo canine open healing GBR study

We studied GBR therapy to verify that the SIOM induced a symbiotic regeneration environment in a complex and challenging cases. As shown in Fig. 5a, the open healing of the SIOM-assisted GBR process was investigated in a medium-sized canine model. This canine pre-molar model offers excellent anatomical similarity to the human alveolar region. During the healing periods, the CM or SIOM was exposed to the oral cavity and came into contact with saliva and gingival crevicular fluid (GCF)[53]. A balanced and diverse oral environment, *i.e.*, symbiosis, promotes physiological healing, while the absence of symbiosis precipitates an environment that interrupts the healing process[54]. This scenario is critical in the early stages when the bacterial interactions feature a more significant impact. Hence, within 48 h of post-GBR surgery, we obtained microbiome information from the supra-membranous region and monitored the surgical wound closure percentage (Fig. 5b)[34]. Fig. 5c describes the sample gathering period, with the GBR surgery point (Sx) serving as the standard. Additionally, GCF was collected from pristine gingival crevices as a healthy symbiosis reference 24 h before GBR surgery (Sx-24). The free gingival

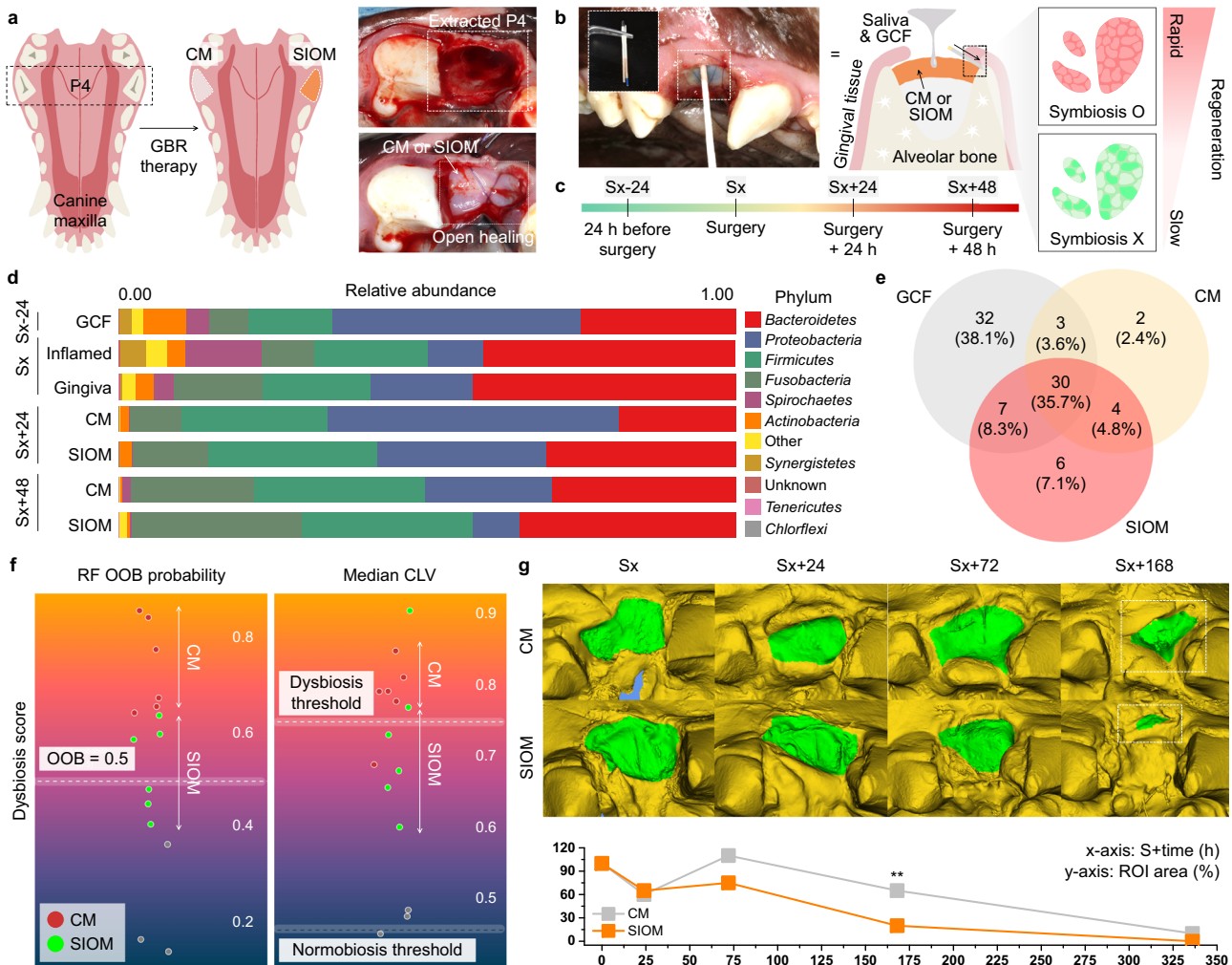

**Fig. 5 | Investigation of open healing during GBR therapy with the SIOM in vivo. a** Illustration of the split-mouth experiment protocol. The third premolar (P4) of a mongrel (medium-sized dog) was hemi-sectioned and extracted. The CM and SIOM were placed to cover the defect and secured with eight sutures. **b, c** Periodic sampling protocol and timeline. The sterile paper was used to acquire the microbiome samples. The gingival crevicular fluid (GCF) and saliva dynamically interacted with the CM and SIOM as the peripheral gingival tissues underwent healing. **d** Taxonomic relative abundance at the phylum level presented across the sampling timeline. **e** Core microbiota distribution between the GCF, CM, and SIOM, observed at the species level resolution. **f** Dysbiosis scores presented using out-of-bag (OOB) probability scores and median community-level variations (CLVs). The dysbiosis threshold was established with pre-surgical GCF samples on a scale of 0 to 1, where 1 indicates the maximum dysbiosis score. **g** Healing rate of the post-surgical wound ($n = 7$, RM ANOVA, $p = 0.002$) compared by identifying the region of interest (ROI; y-axis) and its reduction with time (x-axis). Source data are provided as a Source Data file. $^*p < 0.05$, $^{**}p < 0.01$, $^{***}p < 0.001$.

tissue (Gingiva) and the sub-apically inflamed granular tissue (Inflamed; pathogenic reference) were acquired at Sx. These GCF, Gingiva, and Inflamed samples were further subjected to microbiome analysis. The multinomial logistic-normal model revealed no significant effect of time on microbial variance.

Based on posterior prior analysis, differential ranking analysis was performed to compare the microbiome information for each group[55]. In particular, comparisons were made between the baseline GCF samples and the CM and SIOM. Initially, the GCF showed a grouping pattern for species richness against sequence sample size. Contrarily, the SIOM showed a change in species richness, becoming more distinct at Sx+48 (Supplementary Fig. 18). The alpha diversity metric at the sequence level evidenced a reduction in diversity for the SIOM compared to the CM, implying fewer unique operational taxonomic units in the SIOM (Supplementary Fig. 19). ANCOM-BC analysis suggested that both the CM and SIOM showed differentially abundant phylum-level expression when compared to the GCF (Supplementary Fig. 20). The beta-coefficient magnitude and its direction in Supplementary Table 3 indicate the degree of differential abundance with larger absolute values representing stronger differential abundance.

Although CM and SIOM exhibited an increase in the relative abundance of the *Fusobacteria* content, SIOM presented a resistance towards *Proteobacteria* expression, with a progressive reduction up to the Sx+48 point (Fig. 5d). An increase of *Proteobacteria* expression in lesions has been observed when progressing to periodontitis in dogs[56]. Accordingly, the monitored progressive reduction in the SIOM means its differential resistance to *Proteobacteria*. Furthermore, the SIOM featured an improved expression of the core and other microbiota with an overall 8.3% overlap with GCF (Fig. 5e). Moreover, the overlapping species were notably higher at 10.8% when resolving the SIOM's identified taxa at Sx+48 point (Supplementary Fig. 21). In contrast, the CM had an overall 3.6% overlap, which was particularly lower (2.2%) at the same time point. Thus, the SIOM was capable of resembling the native oral environment during the GBR therapy.

Two statistical indexes were used to quantify the symbiosis-inducing performance of the SIOM: OOB predicted probability and CLVs (Fig. 5f). OOB and CLV (neighborhood classification method by community) are scoring indexes for disturbed symbiosis, i.e., dysbiosis[47]. The dysbiosis threshold was defined from the GCF. The SIOM had a low to medium OOB score, whereas the CM had a predominantly high dysbiosis score (>0.6). The SIOM showed primary distribution under the dysbiosis threshold (0.74) in the median CLV score. Both results indicate that the SIOM resisted dysbiosis induction more effectively than the CM. In other words, osteo-mimetic SIOM positively promotes a microbiome community desirable for regeneration. Although the SIOM's enhanced resistance towards dysbiosis was proved in vivo, there is a limited similarity (16.4%) of the bacterial taxa between canines and humans[57]. Given this limitation, it would be premature to explore a mechanistic rationale with the currently available data.

Figure 5g showed the rapid gingival tissue regeneration under the SIOM-triggered well-established microbial symbiosis. Notably, the SIOM presented a marked reduction in the early surgical wound size within Sx+72. Furthermore, it achieved near complete healing in less than one week of GBR therapy (Sx+168). At Sx+168, the SIOM and CM exhibited significant differences. When the CM was deployed, the healing efficiency of the SIOM at Sx+168 took twice as long (Sx+336) to achieve. Although HAP has outstanding physio-functions in promoting hard tissue regeneration and cellular differentiation[58,59], its relevance to soft tissue regeneration is rarely reported. On the other hand, the importance of microbiota composition in the systemic regeneration process has been gradually revealed[60]. In this context, Fig. 5g implies that the SIOM-based symbiosis induction significantly helps an expeditious healing response during GBR therapy.

## In vivo canine closed healing under the inflammatory-challenged condition

Osteo-immunogenic assays provide information on the healing progress of a complex periodontal defect. During GBR therapy, it is important to create a space that is separate from other tissues and to use an occlusive membrane to preserve that space for closed healing[61]. However, during healing periods, prolonged inflammation causes connective tissue hyperplasia, which deforms the occlusive membrane and compromises the healing space[11]. The inflammation is generally caused by bacterial metabolites, such as lipopolysaccharides, which trigger cytokine expression in neutrophils, macrophages, and B-cell immune cells[62,63]. Accordingly, the inflammation would limit the amount of bone formation in the closed healing zone.

In this study, an inflammatory-challenged closed healing case study further confirmed the symbiosis-inducing capacity of the SIOM in vivo. Chronic periodontal and endodontic lesions were induced for 9 weeks to prepare the inflammatory-challenged environment in soft and hard tissues, which featured severe periodontitis and peri-apical granulomatous lesions (Fig. 6a). Following infected roots extraction, bone defect was treated with synthetic bone substitute with either SIOM or CM, and subsequently covered with the inflamed gingival tissue (Fig. 6b). A contralateral socket was prepared as a comparison reference and denoted as blank. Blank group means the untreated and uninflamed region under spontaneous healing without pin fixation. The study on the closed healing model was limited to immuno-osteogenic bioactivities, and invasive sample collection for microbiome analysis was impossible.

To label inflammatory cell infiltrates, we used MPO for neutrophils and macrophages, CD86 for M1 macrophages, and CD20 for B cells (Fig. 6c–e). The immunohistochemical staining results revealed that MPO increased dramatically in the CM group. In contrast, the SIOM demonstrated strong resistance to neutrophil infiltration, indicating an early alleviation of the acute inflammatory phase (Fig. 6c). Furthermore, we observed that the SIOM group exhibited a 3.3-fold reduction in CD86$^+$ expression when compared to the CM group. CD86$^+$ expression is associated with a pro-inflammatory (M1 macrophage) response. Accordingly, this reduced biomarker expression suggests the occurrence of early macrophage polarization, which is an essential step in tissue healing (Fig. 6d)[64,65]. Moreover, the reduced M1 macrophage expression in the SIOM-treated defect indicated an improved resistance to foreign body reactions. Clinically, peri-implant and periodontal diseases have been recognized through a high B cell density, which is associated with acute exacerbation and aggressive inflammatory lesions[66]. Hence, the multifold reduction in CD20$^+$ observed with SIOM further validates the decreased foreign body reaction (Fig. 6e). In summary, SIOM-assisted GBR therapy can prevent inflammatory responses from exacerbating even when the surrounding environment makes the healing region prone to inflammation.

While an initial inflammatory response is necessary for healing, an optimized immune response can lead to a smoother transition to the anabolic regenerative stage. In our study, the SIOM resulted in excellent biological integration, as evidenced by the micro-CT (Fig. 6f) and micromorphological analyses (Fig. 6g). We observed that the NBV of SIOM significantly improved when compared to that of the blank group. Additionally, the Masson's trichrome-stained histological sections revealed that the NBA in the SIOM group significantly increased by ~67% and 64% in comparison to that in the CM and blank, respectively (Fig. 6h) groups. The entire above results on MPO, CD86, CD20, NBV, and NBA confirmed a symbiotically integrating GBR therapy driven by SIOM.

## Discussion

One of the challenges in GBR therapy for complex alveolar bone defects is the occlusive membrane that guides the suitable regenerative process. Herein, a tooth-enamel-inspired SIOM is proposed as a

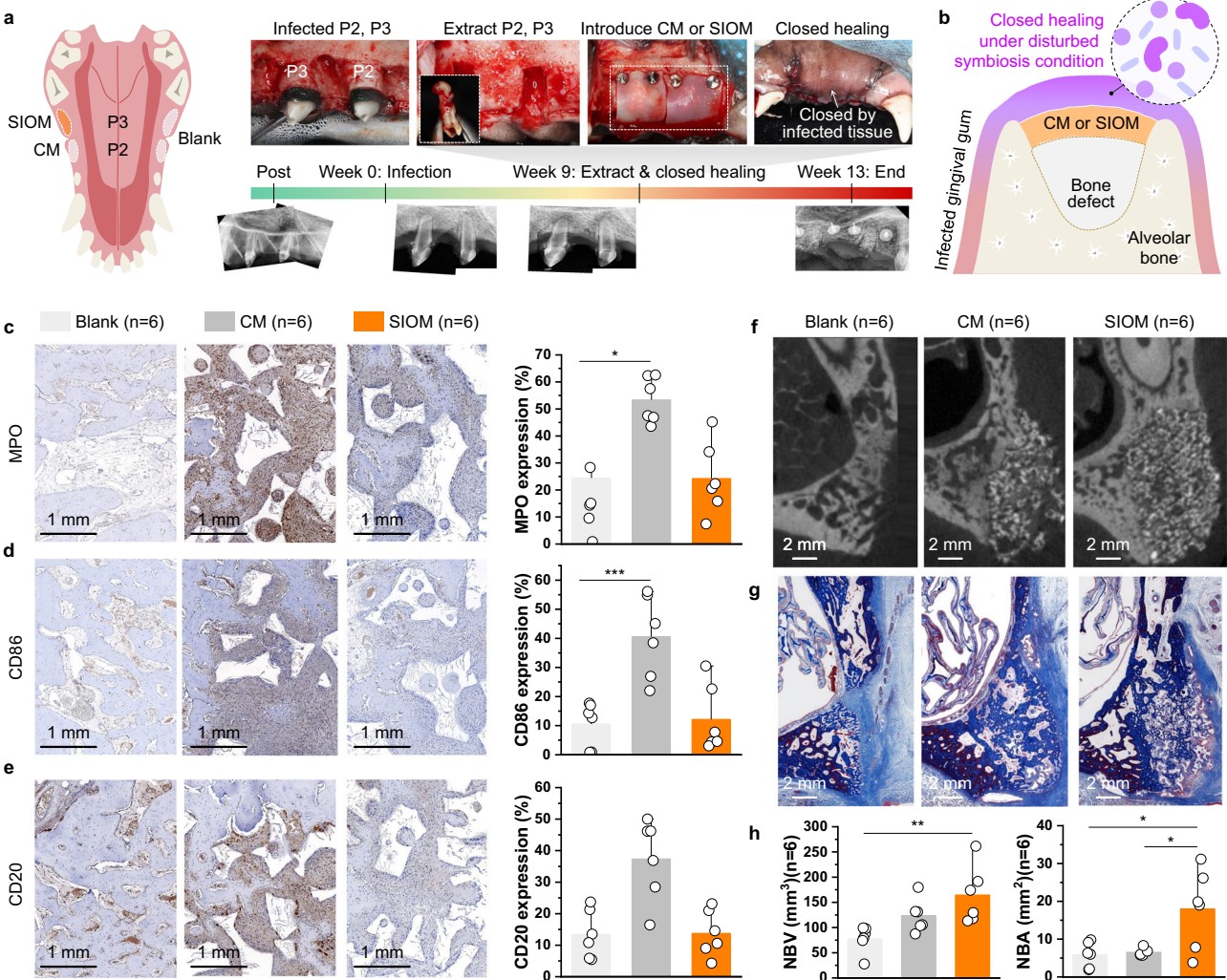

**Fig. 6 | Immuno-osteogenic bioactivities during closed healing. a** Inflammatory-challenged defect model and the intervention protocol. The second and third maxillary premolar (P2 and P3) with ligature-induced periodontitis (9 weeks) exhibits apical granulomatous lesions. The subsequent extraction and delivery of the SIOM and CM were followed by pin fixation. A contralateral root was prepared as a blank control reference. **b** After-extraction defects treated with a bone substitute and the occlusive membrane (CM or SIOM), with a complete flap approximation and surgical wound closure. **c**–**e** Immuno-histological profile and corresponding stained intensity percentages of pro-inflammatory cytokines with specificity for **c** neutrophils (myeloperoxidase; MPO, mean ± SD, $n = 6$, ANOVA, $p = 0.048$), **d** M1 macrophages (CD86, mean ± SD, $n = 6$, ANOVA, $p = 0.001$), and **e** mature B cells (CD20, mean ± SD, $n = 6$). **f** Micro-CT images and **g** Masson's trichrome stained histological photomicrographs obtained after 4 weeks of closed GBR therapy. **h** New bone volume (NBV, mean ± SD, $n = 6$, Kruskal, $p = 0.004$) and new bone area (NBA, mean ± SD, $n = 6$, ANOVA, $p = 0.012$ for blank *vs*. SIOM, $p = 0.024$ for CM *vs*. SIOM) measurements from micro-CT and histological photomicrographs, respectively. n indicates the number of biologically independent animals examined. Source data are provided as a Source Data file. $^{*}p < 0.05$, $^{**}p < 0.01$, $^{***}p < 0.001$.

promising solution for GBR therapy. The SIOM was prepared through a multiple nucleation biomineralization that mimics osteological structures. The SIOM has a symbiosis-inducing effect that balances interactions between microbes and cells. Notably, the SIOM exhibited GBR-therapy-favorable microbial interaction although it was exposed to both a multispecies sample and an inflammatorily challenged condition. Thus, it can prevent negative inflammatory responses, reduce dysbiosis, and enhance cellular activity. Finally, expeditious and reliable GBR therapy is achieved. The symbiosis-inducing effect of the SIOM can reduce the need for additional systemic drugs. Moreover, the SIOM has excellent tractability and space-forming abilities that reduce the probability of premature membrane loss and insufficient new bone growth. In summary, the SIOM creates a highly physiological healing environment that fosters optimal bone regeneration, enabling the possibility of next-generation GBR.

Recent works prove that artificial transition in microbiome conditions can alleviate dysbiotic diseases; for instance, obesity[67], diabetes[68], cancer[69], and metabolic syndrome[70]. Thus, worldwide efforts are in progress to understand and regulate the human microbiome; for instance, the Human Microbiome Project (HMP) in the USA and the Metagenomics of the Human Intestinal Tract (MetaHIT) in the EU. However, the reported microbiome-targeting therapeutic options, such as diet, pro- and prebiotic intervention, bariatric surgery, and fecal transplant, are generally applicable to the intestinal microbiome[71]. The oral microbiome is the second most diverse community in the human body and significantly influences the local and systemic health equally to the intestinal microbiome[13,72,73]. Hence, the oral microbiome should be taken into account. However, oral microbe-related material technologies are still focused on total prevention of bacteria accumulation, for instance, antifouling[49,74] and bactericidal effects[75–77].

## Methods

### Inclusion & ethics

The human saliva was obtained following the ethical principles of the 64th World Medical Association Declaration of Helsinki and

procedures approved by the institutional review board of the Yonsei University Dental Hospital (Republic of Korea) (Approval No.2-2019-0049). The consent was obtained from all participants before donating saliva. All the animal experiments were approved by the Institutional Animal Care and Use Committee at Yonsei Medical Center, Seoul, Korea (Approval No.2022-0086).

## Preparation of ZM and SIOM

All reagents were acquired from the producer (Sigma Aldrich, USA). Clinical grade type I collagen membranes (CM; dry thickness = 300 μm) were obtained from the manufacturer (GENOSS, Korea). CM was treated with $O_2$ plasma for 2 min and 10% (w/w) benzophenone dissolved in ethanol for 3 min. After washing with deionized water, CM was immersed in precursor and exposed under 365 nm UV for 3 h. The ZM precursor was 22.5wt% [2-(methacryloyloxy)ethyl]dimethyl-(3-sulfopropyl)ammonium hydroxide, 1.50wt% alginic acid, 0.75wt% 2-hydroxy-4′-(2-hydroxyethoxy)-2-methylpropiophenone in deionized water. The precursors for NM, PM, and ZM-2 were prepared using 22.5wt% sodium 4-vinylbenzenesulfnoate, diallydimethylammonium chloride, and 2-methacryloyloxyethyl phosphorylcholine with identical reagents recipe. After UV reaction, the samples were moved into 0.1 M $CaCl_2$ to ionically crosslink the alginic acid for 2 h. Then, ZM was prepared after a vigorous wash with deionized water. ZM was stabilized under deionized water for 1 d and transferred to 40 mM Tris buffer (pH 8.5). The biomineralization of ZM was performed by repetitively immersing in 0.3 M $K_2HPO_4$ and 0.5 M $CaCl_2$ solutions. This protocol started with 0.3 M $K_2HPO_4$ and finished with 0.5 M $CaCl_2$. Every immersion step was progressed for 1 min, and the deionized water wash (10 s) was followed before moving to the counter ion medium. After ten times of $K_2HPO_4$-$CaCl_2$ cycles, the sample was incubated in 0.5 M $CaCl_2$ for 1 d under 37 °C. Finally, the biomineralized SIOM was stored in deionized water until its utilization.

## Characterization of ZM and SIOM

The top view images and ion mapping results were obtained using the field emission scanning electron microscope (JSM-IT-500HR, JEOL, Japan). Raman signals were acquired using Raman spectroscopy (XploRA PLUS, HORIBA, France) with a 10× microscope objective and a 532 nm laser. The laser intensity was about 75 mW. The crystalline silicon peak at 520 $cm^{-1}$ was regarded as the calibration standard. For each measurement, the laser was exposed for 30 s and repeated five times. SAXS and WAXS experiments were performed with 9 A X-ray beamline at the Pohang Light Source (Korea). WAXS measurements were performed with a sample-to-detector distance of 0.2 m and exposure times of 5–10 s. SAXS measurements were performed with a sample-to-detector distance of 2.5 m and exposure times of 10–30 s. The rheological response was monitored under the frequency sweep condition (amplitude 1%, angular frequency 5 rad $s^{-1}$) using the rheometer (MCR 302, Anton Paar, Austria).

## Enzymatic degradation test

0.2 IU $mL^{-1}$ of collagenase type I was prepared in 1× Hank's balanced salt solution with 10 mM $CaCl_2$. 1 × 1 $cm^2$ samples were immersed in 24 well containing 1 mL of collagenase media. Subsequently, the samples were incubated at 37 °C for enzymatic degradation. The thickness of the residual membrane was measured at each time point. The shear moduli of MC and SIOM were measured during the biodegradation in situ. The frequency sweep mode followed the protocol above.

## In vitro cell experiments

**Cell culture.** BMSCs (American Type Culture Collection (ATCC), PCS-500-012) were cultured in Minimum Essential Medium, Alpha Modification (Alpha MEM, WELGENE, LM 008-01) supplemented with 10% (v/v) fetal bovine sera (FBS, WELGENE, S 001-07) and 1% (v/v) penicillin-

streptomycin (Gibco, 15140122). Here, BMSCs with #5-8 passages were employed for in vitro experiments. BMSCs were cultured in growth media and osteogenic differentiation media for 2 weeks. Osteogenic differentiation medium was prepared by adding 10 mM β-glycerophosphate (Sigma-Aldrich, G5422-25G) and 50 μg/mL ascorbic acid (Sigma-Aldrich, A5960-25G) to Alpha MEM culture media containing 10% (v/v) FBS. BMSCs were cultured in $CO_2$ incubator (Thermo Fisher Scientific, BB 150) at 37 °C and 5% $CO_2$.

## Quantitative real-time polymerase chain reaction

Total RNA was extracted using RNeasy Mini Kit (Qiagen, 74140). cDNA was synthesized by SuperScript III First-Strand with oligo-dT primer (Invitrogen, 18080-051) following the manufacturer's instruction. SYBR Green PCR master mix (Applied Biosystems, Foster City, 4309155) and QuantStudio 3 Real-time PCR (Life technologies, A28131) were used for the qPCR assays. The cycling parameters of the PCR experiment were: 95 °C for 15 min, 95 °C for 10 s, 60 °C for 15 s, and 72 °C for 30 s for 40 cycles, followed by 72 °C for 10 min and 95 °C for 10 s. The relative gene expression level was quantified by the delta Ct method. The primers used in this study were listed in Supplementary Table 4.

## In vitro microbiological assays

**Sample preparation.** Human-saliva-derived biofilm analyses were carried out according to previously established methods[78]. The saliva was obtained following the ethical principles of the 64th World Medical Association Declaration of Helsinki and procedures approved by the institutional review board of the Yonsei University Dental Hospital (Republic of Korea) (2-2019-0049). The consent was obtained from all participants before donating saliva. The self-reported gender and/or sex was not a study variate, and recruitment was independent of the same. The human saliva from 6 human donors (male = 3, female = 3) was collected. The human salivary sample data collected from 6 healthy donors was pooled before the incubation with the occlusive membranes, and individual-level data was not obtained. Briefly, the unstimulated salivary samples were mixed in equal proportions and then diluted to 30% in sterile glycerol and stored at −80 °C. McBain medium was prepared to simulate the saliva environment for cultivating biofilm models. Human saliva was cultured in McBain medium for 24 h, dropped onto specimens (1.5 mL), and then cultured at 37 °C for 48 h. During this period, the medium was replaced with fresh medium per 8, 16, and 24 h.

## 16S rRNA sequencing assay

DNA sequencing was performed to identify the biofilm composition on specimens, as previously described[79]. Briefly, the bacterial DNA was extracted from the samples using the DNeasy PowerSoil Kit following the manufacturer's instructions. Each sequenced sample was prepared according to the Illumina 16S rRNA sequencing library protocols. PicoGreen and VICTOR Nivo were employed to evaluate the quality and quantity of DNA. The 16S rRNA genes were then amplified using 16S V3-V4 primers, with an additional limited-cycle amplification step for the inclusion of multiplexing indices and Illumina sequencing adapters. Subsequently, the resulting products underwent normalization and pooling, and size confirmations, before being subjected to sequencing on the MiSeq™ platform.

For the amplicon sequencing, the Illumina Miseq Sequencing System from Illumina, USA was utilized. To ensure data integrity, a quality check on the raw reads, filtering out low-quality reads with a score of less than 25 was performed. The Divisive Amplicon Denoising Algorithm 2 (DADA2 version 1.26)[80] was applied within R Studio (R version 4.2), to conduct the assembly of paired-end reads and the subsequent assignment of these reads to amplicon sequence variants. The unique 16S rRNA reads were isolated at similarity threshold of 97%, and allocated to taxonomy based on the EzBioCloud 16S rRNA

database[81]. The raw sequencing data generated in this study data have been submitted to the NCBI BioProject database under accession number PRJNA981675.

## In vivo experiments

**Animals.** A total of 6 male mongrel dogs, 10–14 months of age, weighing 25–30 kg (CRONEX, Suwon, Korea) were included in the present study. In order to avoid experimental differences caused by animal sex, animals of the same sex were used. Animals were housed in separate cages under standard laboratory conditions with free access to water and diet. Experiments were performed in the socket of the maxillary premolar on both sides of each animal[82,83]. CM or SIOM was introduced experimental site of each animal. 0.5 cc of particle-type bone grafts (Osteon III; Genoss, Korea) made up of porous 60% HAP and 40% tricalcium phosphate with a particle size of 0.5–1.0 mm was deployed. Groups were randomly assigned. Animals were sacrificed after a healing period of 4 weeks in the experiment.

## Surgical procedure of GBR therapy

All surgical procedures were performed in a sterilized operating room with the mongrel dogs under general anesthesia induced by medetomidine (0.75 mg/kg, i.m.; Tomidin, Provet Veterinary Products, Istanbul, Turkey) and alfaxalone (2 mg/kg, i.v.; Jurox, Rutherford, NSW, Australia), and maintained using isoflurane (Forane, Choongwae Pharmaceutical, Seoul, South Korea) inhalation. Local infiltration anesthesia with 2% lidocaine HCl with epinephrine 1:100,000 (Kwang Myung Pharm, Seoul, Korea) was used at the intraoral surgical sites. Scaling and plaque control were performed before surgery. GBR procedures were done on both experimental models of the extraction socket and infected alveolar ridge.

For extraction socket model, the crevicular incision was performed along the fourth maxillary premolar (P4), followed by a hemisection to extract the distal root. The pulp tissue was removed from the pulp chamber of the remaining mesial root and then sealed with calcium hydroxide paste (Dycal, Dentsply Caulk, Milford, Delaware). After filling with the extraction socket with synthetic bone substitutes (Osteon III; Genoss, Suwon, Korea), CM and SIOM were carefully applied over the extraction socket entrance. The membranes were secured with resorbable sutures (4-0 glyconate monofilament, Monosyn, B Braun, Tuttlingen, Germany) and primary closure was not achieved for open wound healing.

For infected ridge augmentation model, combined endodontic and periodontal lesions were induced in maxillary premolar area. After flap elevation, distal roots of the 2nd and 3rd premolars were carefully removed after hemisection, and then the buccal bone was removed to create a narrow alveolar ridge. And for the remaining mesial roots, plaque suspension was applied in the pulp chamber without root canal treatment and filled with temporary restorative material. Silk ligature was immediately applied to the cemento-enamel junction to induce periodontal lesion. After 9 weeks of healing of interrupted oral hygiene, alveolar bone loss due to endodontic and periodontal lesion was confirmed on periapical radiographs. After extracion of infected mesial roots, inflamed soft tissue was removed and the, GBR procedure was done with CM and SIOM. Buccal flaps were advanced coronally and primary closure was attempted with resorbable sutures (4-0 glyconate monofilament, Monosyn®, B. Braun, Tuttlingen, Germany).

## Microbiome sequencing and ecological indexing from in vivo open healing model

The samples were bilaterally collected in relation to P4 from the gingival crevicular region and from above the surface of CM and SIOM. Entire samples were collected with the help of sterile paper points of the same dimension[43]. For consistency of sample collection, each paper point was allowed to absorb the exudate for up to 30 s or up to one-half the length. The samples were collected 24 h before the extraction and immediately after the membrane insertion. Additional samples were collected at 24 h intervals during the healing period for 2 d. The bilateral samples before membrane placement were pooled to minimize variation in baseline samples. A regular feeding schedule was followed to eliminate dietary changes as a factor. Samples were collected at least 12 h after the last meal and without any additional oral-prophylaxis measures during the assessment period. This protocol was done to ensure repeatability and quality control in the sampling protocol. The collected specimens were filtered to remove bacteria and stored for sequence processing and analysis using the same methods as in the in vitro analysis. 16S rRNA Amplicon sequencing of V3 to V4 were analyzed using Bakt 341F-805R target region. The best BLAST hit in the NCBI_16S_20211127(BLAST) was chosen as the reference for each representative sequence. The raw sequencing data generated in this study data have been submitted to the NCBI BioProject database under accession number PRJNA981675.

## Gingival soft tissue profile analysis

Using an intraoral scanner (Medit i500), the digital impressions of the extraction site were taken up to 2 weeks after surgery. Because the employed intraoral scanner is operated through video-type scanning based on a triangulation technique, it allowed the expeditious acquisition of three-dimensional images from canine intraoral tissues. Then, the scanned images were imported into a digital imaging software program (SMOP; Swissmeda, Switzerland) to define a region of interest (ROI) where area change in the exposed surgical wound area (gingival tissue) was measured.

## Radiographic analysis

Radiographic images were scanned using a micro-CT system at resolutions in uM (using kV and uA), acquired in Digital Imaging and Communications in Medicine (DICOM) file, and subjected to 3D morphological analysis with computer software (OnDemand3D; Cybermed, Korea) to be observed. ROI was set and analyzed inside the extraction socket. The identical threshold value was applied to ROI. In detail, the threshold values ranged from 140 to 255 for residual bone grafts and 83 to 139 for new bone.

## Histomorphometric and immunohistochemical analyses

Tissue block specimens were cut in half based on the most central region of the defect, histological preparation was performed on one side, and immunohistochemical staining was done on the other side. After rinsing the specimens for histologic preparation fixed in 10% formalin, the sections were decalcified in 5% formic acid for 14 days and embedded in paraffin and stained with Masson trichrome. The stained slides were digitally scanned with software (Panoramic 250 Flash III; 3DHISTECH, Hungary) at ×200 magnification and observed in a case viewer (3DHISTECH, Hungary). The images were subjected to histomorphometric analysis through Photoshop software (Adobe, USA).

The tissue blocks for immunohistochemistry were decalcified with EDTA (12.5%, pH = 7) for 4 months. Once fully decalcified, these pieces were embedded in paraffin and cut to obtain 7 µm-thick sections that were used for immunohistochemical staining. The staining was done with the Master Polymer Plus Detection System kit that uses a secondary antibody, on to a base of micropolymers containing the rabbit and mouse monoclonal and polyclonal primary antibodies, and as visualization solution uses DAB (3,3'-diaminobenzidine). All obtained images were standardized by using Adobe Photoshop software (Adobe Inc, San José) in terms of brightness, contrast, and color tone. The stained intensity (%) of MPO, CD86 and CD20 was quantitatively measured using the IHC profiler and Image J (NIH, Bethesda)[84].

Below is the information on the used antibodies. MPO; Supplier name: Antibodies; Catalog number: ABIN5013150; Clone name: 1F10; Applicable: IHC; Dilution: 1:200, CD86; Supplier name: Antibodies; Catalog number: ABIN736701; Clone name: BU63; Applicable: IHC; Dilution: 1:200, CD20; Supplier name: Novus Biologicals; Catalog number: NBP2-70362H; Clone name: OTI4B3; Applicable: IHC; Dilution: 1:200

### Statistics analysis

For two group comparisons, a two-tailed students *t*-test was used. For multiple group comparisons, ANOVA was used at one-time point. RM ANOVA was used for between-group tests at each time point. Statistical significance was set at 5%. During in vivo experiments, all measurements were evaluated by experienced examiners blinded to group assignment. Software (SPSS version 23; IBM, Armonk, NY, USA) was used for statistical analysis.

### Reporting summary

Further information on research design is available in the Nature Portfolio Reporting Summary linked to this article.

## Data availability

The raw sequencing data generated in this study data have been submitted to the NCBI BioProject database under accession number "PRJNA981675". All other data supporting the findings of this study are available within the article and its supplementary files. Any additional requests for information can be directed to, and will be fulfilled by, the corresponding authors. Source data are provided with this paper.

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

## Acknowledgements

SAXS and WAXS measurements were performed on the 9A beamline at the Pohang Accelerator Laboratory, Korea. This research was supported by (1) National Research Foundation of Korea (NRF) grant funded by the Korea government(MSIT)(No. 2021R1A4A3030268, S.L.), (2) Korea Research Institute of defense Technology planning and advancement (KRIT) - Grant funded by Defense Acquisition Program Administration(DAPA) (KRIT-CT-21-034, J.H.), (3) international cooperation program managed by the National Research Foundation of Korea (NRF-2022K2A9A1A0609182711, J.H.), (4) Korea Evaluation Institute of Industrial Technology (KEIT) grant funded by the Korea Government (MOTIE) (20023781, S.H.C.), (5) the Institute for Project-Y Seed Grant of 2023 (S.H.C.), (6) National Research Foundation of Korea (NRF) grant funded by the Korea government (MSIT) (No. RS-2023-00217709, S.H.C.), (7) a grant of the Korea Health Technology R&D Project through the Korea Health Industry Development Institute(KHIDI), funded by the Ministry of Health & Welfare, Republic of Korea (grantnumber: HI22C1609, J.K.C.), and (8) Bio & Technology Development Program of the National Research Foundation (NRF) & funded by the Korean Goverment (MSIT) (No. 2022M3A9F3016364, J.K.C.).

## Author contributions

W.C., U.M., S.H.C. and J.H. conceived the presented idea and performed experiments. J.Y.Na., J.Y.P., and J.K.C. designed and conducted the in vivo experiments. J.Y. K. carried out the human saliva-derived in vitro studies. T.J. and D.Y.R. performed the X-ray scattering analysis. J.W.J. and J.M.K. contributed to the in vitro BMSC tests. M.C., S.J., M.L., and J.S.K. analyzed the data. W.G.K., S.L., P.T.J.H., K.J.L., and U.W.J. reviewed the manuscript.

## Competing interests

The authors declare no competing interests.
