## [Peer review file · Nature Communications]

REVIEWER COMMENTS

Reviewer #1 (Remarks to the Author):

Thanks for the article and I have the following comments:

1. p.4 L49-51 "The annual economic burden of inflammatory periodontal diseases is estimated to be over \$150B..." is incorrect. The original article was mentioning the figures in Europe and US separately in different amount, whereas the total is not (USD?) 150B... Please check correctly and amend.
2. So, the purpose of your membrane is to regenerate the bone defect in dental implant, or simply for bone loss induced by periodontal diseases? The Figures 1 and S3 are not sync. I guess these two diseases may not have the same root cause (although bacteria play a role) so please explicitly mention the application. I was quite confused in the Main introduction. This two scenario would also need different animal models.
3. Yo have used ex vivo human saliva model for the microbiome evaluation, but you have tested using the mongrels dog animal. They should have different microbiota and distribution. How to make these two models comparable with each other? Any transgenic mongrels model or other model that can spectacularly representative for human oral microbiome condition?
4. How can you distinguish the effects from the membrane vs bone graft HAP/TCP material?
5. How well would the intraoral scanner can establish for scanning dog teeth? Normally the scanner is optimised for human dentition.
6. I understand the membrane can be stretched quite a lot, so under the stretched condition, would the change of porosity or pore size affect the outcome?
7. If the wound is being attacked by other bacteria, the symbiosis may be destroyed. How to proof your membrane is able to resist this kind of situation ?
8. what is the degradation rate of your SIOM membrane? it looks like this is very stable...

Reviewer #2 (Remarks to the Author):

Symbiotically Integrating Occlusive Membrane for Guided Regeneration of Inflammatory-Challenged Complex Tissue Defects, by Choi et al uses two assays to examining bacteria profiles on different membranes used to expedite healing and bone regeneration in the oral cavity. The ex vivo assay with human saliva is not really an ex vivo assay as samples are frozen in 30% glycerol prior to study, which kills many of the taxa. However, it is an interesting assay used to determine which oral bacteria will adhere and thrive on the surfaces of the various tested materials use to promote bone healing. This is done with the idea that some bacteria will interfere with the healing process. The second oral microbiome assay is done in vivo and relies on healing at a tooth extraction site in dogs with the different test materials used to promote healing. Bacteria samples are collected from crevicular fluid and from above CM and SIOM surface with paper points

Ex vivo or in vitro assay:

Figure 4 would have more value if the taxa identification was done on the species level. For example Porphyromonas gingivalis is a periodontal pathogen while other species of Porphyromonas in the oral cavity are not. Given the usage of human saliva for the ex vivo study I would suggest not use NCBI_16S_2021112 as the reference library for taxa identification and instead use a library restricted to human associated bacteria or that of the oral cavity. Or maybe sequence a different part of 16S rRNA gene.

301 Furthermore, in specific periodontally related indicator species, the pathogenic Porphyromonas and Fusobacterium genera were observed in 302 the PC and CM groups only (Figure 4h).

This has more meaning when taxa are identified on species level.

Figure S11. Core abundance index (CAI). CAI provides a useful metric for assessing the 265 health of microbial ecology by focusing on the most important species and their abundance. It

266 represents the averaged abundances of core species identified (>80% prevalence). A high CAI
267 indicates a well-functioning microenvironment with adequate populations of key species,
268 whereas a low CAI may indicate stress or disturbance.

It is hard to know the point of this figure. It seems to indicate what the other figures show, that with usage of SIOM or ZM the average abundance of the core taxa is much higher. Isn't this just another way of saying that with the usage of these 2 materials there is much less variety of taxa present, so the ones present are at higher relative abundance. Contrary to what is suggested in the figure legend above loss of diversity is often associated with disease, though in periodontal disease that may be controversial. It is interesting that the different materials used have different taxa adhering but it is premature to claim that that is because fewer pathogens are present, though the authors can certainly speculate about it. Without species level identification they have not even shown association with periopathogens.

The authors need to discuss more whether this reduction in diversity is seen in the in vivo studies using the canine model. Figure 5 d is only on the phylum level. Is the alpha diversity lower on the DNA sequence level for the SIOM versus CM membranes as the ex vivo study indicated?

There is nothing wrong with the authors suggestion that the SIOM membrane works well due to lack of inflammation associated bacteria. However with the large decreases in taxa number, approx. 7x, it is hard to identify specific bacteria taxa responsible for differences if they indeed are causative.

Table S4 Please have columns with relative levels of taxa on CM and SIOM and maybe the FDR value for each comparison. Additionally define Beta-coefficient(w). There needs to be more of an effort to make the results understandable.

Line 317

Dysbiosis scores presented using out-of-bag (OOB) probability scores and median community-level variations (CLVs). Presumably a healthy microbiome is based on what is present prior to surgery. The reference for dysbiosis, Wei, S., Bahl, M.I., Baunwall, S.M.D., Hvas, C.L. & Licht, T.R. Determining gut microbial dysbiosis: a review of applied indexes for assessment of intestinal microbiota⁵⁴² imbalances. *Appl. Environ. Microbiol.* 87, e00395-00321 (2021) does not explain CLV. Presumably what the authors are saying is that during healing with SIOM the variation of the bacteria taxa seen are most similar to the healthy state. It is a little puzzling why Fusobacteria earlier stated as being linked to disease would increase with SIOM. More effort needs to be made to explain differences in in vitro (ex vivo) and in vivo results in regard to microbiome.

Line 268

to study the microbiome development at the SIOM, we used metagenomic sequencing of co-cultured salivary extracts from six healthy adults (Figure 4c
Please use the term 16S rRNA sequencing. Metagenomic sequencing is usually reserved for shotgun sequencing of multiple organisms.

Line 278 Thus, a settled microbiome community with a unilaterally low F/B scan (impedes- delete). (May Impede-add) normal healing through a chronic expression of pro-inflammatory cytokine. Speculation is stated as fact.

Line 300 Furthermore, in specific periodontally related indicator species, the pathogenic Porphyromonas and Fusobacterium genera were observed in the PC and CM groups only (Figure 4h).

Need to identify these bacteria on species level as for example not all oral Porphyromonas that are in the periodontium or are pathogenic.

Line 360

An established microbial symbiosis results in an expeditious regeneration response. As proof, we observed an improvement in the open healing rate of the gingival tissue during GBR therapy with the SIOM (Figure 5g).

The result support the conclusion but is by no means proof of it. The claim needs to be toned down.

Major revision required.

Reviewer #3 (Remarks to the Author):

The authors present a significant advancement in the field of oral tissue engineering with their development of a symbiotically integrating occlusive membrane (SIOM). This innovative membrane is designed to promote the regeneration of a biologically durable tooth-like environment, particularly at the interface of closed healing and open healing regions. The study investigates the potential of SIOM biomineralization and enzymatic durability, as well as its ability to induce symbiosis, *ex vivo* and *in vivo*. An important aspect that warrants further exploration is the influence and interaction of the oral microbiome with the engineered membranes. Understanding this interaction is crucial for comprehensive analysis and should be considered in future research. The work is interesting and novel, offering promising considerations for more realistic oral soft or hard tissue engineering approaches. The authors provide comprehensive information regarding the synthesis and characterizations of the SIOM membrane, demonstrating a strong foundation for their findings. However, to meet the criteria for publication in *Nature Communications*, some clarifications and justifications are necessary. These improvements will further enhance the quality and impact of the research:

- 1) The authors used the rheological assessment to study the SIOM mechanical properties after biomineralization using indicators of storage modulus and $\tan(\delta)$. The $\tan(\delta)$ shows the ratio of lost energy to stored energy during cyclic deformation. How can this information demonstrate the mechanical properties of hard and soft tissue? The proper informative explanation is missing. Please provide more discussion.
- 2) SIOM membrane is the biomineralized zwitterionic membrane (ZM). The authors mentioned that the mechanical properties of a fully covered HAP (mineralized) membrane (which is a ZM membrane) decreased, but SIOM showed higher mechanical properties (Figure S5). How SIOM presented higher mechanical properties than ZM, while SIOM is a biomineralized form of ZM? This feature requires proper justification.
- 3) The oral environment harbors a vast array of bacterial species, numbering over 700. Figure 4 in the study presents data regarding the specific genus under investigation. It would greatly enhance the manuscript if the authors could provide a concise explanation for the rationale behind studying these particular genera. Such information would offer valuable insights into the selection process and help readers understand the significance of focusing on these specific bacterial groups within the context of the research.
- 4) Authors studied osteogenic differentiation using qPCR. However, the duration of the study is missing. By specifying the duration, the authors can effectively demonstrate the extent to which the SIOM promotes osteogenic differentiation and highlight its potential for long-term or short-term tissue regeneration.
- 5) Regarding the discussion of Figure 4f, it will improve the article significantly if the authors correlate data with native oral microbiome diversity and frequency.
- 6) Furthermore, it would greatly enhance the manuscript if the authors could provide clarification and a comprehensive list of the healthy and diseased oral microbiome (specifically ones that were studied in this project). This clarification would facilitate comparisons between the outcomes of the study and existing literature, providing valuable insights into the potential applications and limitations of the SIOM membrane in various oral health contexts.
- 7) According to Figure 5, it is evident that the CM membranes exhibit a core-microbiota distribution that is similar to the gingival crevicular fluid (GCF). This similarity is indeed favorable as it indicates a resemblance to the native oral environment. However, it is important for the authors to clearly justify why the SIOM membrane is superior to the CM membranes, considering that the primary aim was to mimic the native tissue environment. It would be also valuable if the authors could discuss any functional or structural features of the SIOM that directly facilitate the establishment and maintenance

of a healthy oral microbiome. Exploring how the SIOM promotes symbiosis and influences the microbial community in a beneficial manner would further substantiate the claim that it provides a more realistic and advantageous condition for oral tissue engineering.

8) In in vivo canine closed healing study, pins were used to fix CM and SIOM membranes. Pin fixation may cause an additional inflammatory response which is one of the limitations required to be discussed. Pin fixation may also explain the high MPO, CD86, and CD20 expression of the CM membrane in comparison with the Blank (control) group. In addition, The experimental results suggest that the pin fixation method used in the study did not have an adverse impact on the properties of SIOM. SIOM demonstrated similar behavior to the Blank group in terms of the expression of pro-inflammatory cytokines. Providing a strong, detailed and well-supported explanation for the superiority of the SIOM over the CM membranes will address this point of concern.

Minor comment:

1) Line 261-264 needs adequate citations. Line 360 requires citations.

Reviewer #4 (Remarks to the Author):

I co-reviewed this manuscript with one of the reviewers who provided the listed reports as part of the Nature Communications initiative to facilitate training in peer review and appropriate recognition for co-reviewers.

Response Letter to Reviewers

Manuscript ID: NCOMMS-23-24435-T

Title: *Symbiotically Integrating Occlusive Membrane for Guided Regeneration of Inflammatory-Challenged Complex Tissue Defects*

Author: *Woojin Choi† , Utkarsh Mangal† , Ji Yeong Na, Ji-Yeong Kim, Taesuk Jun, Ju Won Jung, Moonhyun Choi, Sungwon Jung, Milae Lee, Jin-Young Park, Du Yeol Ryu, Jin-Man Kim, Jae-Sung Kwon, Won-Gun Koh, Sangmin Lee, Patrick T. J. Hwang, Kee-Joon Lee, Jae-Kook Cha* , Sung-Hwan Choi* , and Jinkee Hong**

The authors would like to express our gratitude for the valuable reviewers' comments. We truly agree with your belief in the potential of our work to bring significant advances in regenerative biomaterial science and attract new readers to ***Nature Communications***. In particular, we sincerely hope that the striking finding of our work (*i.e.*, finely designed materials could lead to diverse microbiome communities within both canine and human models) might break an old paradigm of regenerative research.

Recognizing the merit of the reviewers' comments, we took the opportunity to improve our manuscript entirely. We have diligently addressed each concern as shown below, and carefully refined our manuscript to enhance its clarity and depth. Specifically, we have endeavored to clarify the causality of every discussion because this work dealt with an extensive database of microbiome variation. We strongly believe that the reviewers will sympathize with a considerable improvement in the manuscript quality.

In this response letter, the authors did our earnestness to address the reviewers' comments precisely. The followings are point-by-point responses to the comments. (**Page 00**) correspond to the revised document and prepare to assist the reviewer's tracking of revision.

Again, we sincerely appreciate the tremendous contribution. The authors heartily look forward to the favorable responses.

Reviewer #1

Comment #1

p.4 L49-51 "The annual economic burden of inflammatory periodontal diseases is estimated to be over \$150B..." is incorrect. The original article was mentioning the figures in Europe and US separately in different amount, whereas the total is not (USD?) 150B... Please check correctly and amend.

Comment #2

So, the purpose of your membrane is to regenerate the bone defect in dental implant, or simply for bone loss induced by periodontal diseases? The Figures 1 and S3 are not sync. I guess these two diseases may not have the same root cause (although bacteria play a role) so please explicitly mention the application. I was quite confused in the Main introduction. This two scenario would also need different animal models.

Authors' response to Comment #1-2

We appreciate the helpful comments and completely reflect the reviewer's opinion in the revised manuscript. Firstly, we acknowledge that the original description related to the annual economic burden of inflammatory periodontal diseases was ambiguous. By referring *J. Periodontal* (2022), Box R1 suggests the correct information about the annual periodontitis-related indirect cost and its portion in the gross domestic product per each country.¹

Box R1. Periodontitis-related economic burden in US and Europe

The considerable economic burden of periodontitis is attributed to the implant treatment of missing tooth.¹ For instance, in the United States, the annual indirect cost of periodontitis amounted to \$150B, accounting for 0.73% of gross domestic product. This situation is comparable in Europe regarding that €156B periodontitis-related indirect cost takes up 0.99% of gross domestic product.

Subsequently, below is the summarized answer to the reviewer's comment #2: the symbiotically integrating occlusive membrane (SIOM) is proposed to achieve excellent guided bone regeneration (GBR) therapy. In other words, this work focused on advancing GBR therapy, considering its tremendous potential in regenerating both bone defects in dental implants and bone loss induced by periodontal diseases.

Tooth loss can occur accidentally or by inflammatory periodontal diseases (mainly periodontitis). It causes alveolar atrophy, translating into the periodontal loss of soft and hard tissues.² To this end, GBR treatment is performed on diverse edentulism patients.³ As the reviewer mentioned, GBR therapy enables the regeneration of the partial small alveolar bone defect, which does not require dental crown implantation.⁴ Furthermore, regarding an adequate alveolar bone volume is a prerequisite for successful implant treatment, GBR treatment promotes the satisfactory outcome of tooth implant.⁵ To improve these broad applications of GBR therapy in dentistry, we developed the SIOM to retain the symbiosis and accelerate bone augmentation. Accordingly, we thoroughly investigated the open healing and closed healing scenarios in Figure 5-6, proving the significance of the SIOM in GBR therapy after a tooth missing.

We acknowledge that the original Figure S3 could leave a misunderstanding

about the application field of SIOM. The authors intended to emphasize the tractability of SIOM, enabling the handy operation during dental surgeries. Hence, we revised and updated the related descriptions. Moreover, we removed the periodontitis-related economic burden section, forming a reader's firm focus on GBR therapy.

Changes in manuscript reflecting Comment #1-2

(Page 3)

Tooth loss can occur accidentally or by inflammatory periodontal diseases, primarily periodontitis. It leads to alveolar atrophy, translating into the loss of periodontal soft tissue (gingival gum) and hard tissue (alveolar bone). Because an adequate alveolar bone volume is a prerequisite for successful implant treatment, guided bone regeneration (GBR) therapy has been performed on diverse edentulism patients. For instance, repairing critical size defects using GBR technology has led to the success of dental implant procedures, with over 40% incidence of clinical use.³ Therefore, effective and long-term GBR technology is necessary for the broad dentistry fields. Recently, the satisfactory outcome of GBR has been achieved through advanced occlusive membrane technologies.

(SI, Page 12)

The SIOM is tractable, enabling readily application during dental surgeries.

Comment #3

You have used ex vivo human saliva model for the microbiome evaluation, but you have tested using the mongrels dog animal. They should have different microbiota and distribution. How to make these two models comparable with each other? Any transgenic mongrels model or other model that can spectacularly representative for human oral microbiome condition?

Authors' response to Comment #3

The authors truly thank the reviewer for suggesting meaningful feedback. We sincerely acknowledge that there may be inherent and inevitable variation in the microbiota between the human and canine models, which can exhibit questionable fidelity when studying microbiota-specific signatures. Regarding the transgenic microbiome model, the transgenic rodent was suggested *via* fecal and oral microbiota transplantation; however, only a limited 84.78% taxonomic replication of the genus level patterns was achieved.⁶

We wish to persuade you that we have contemplated and employed the canine model considering the primary goal of this study: developing an advanced occlusive membrane and thoroughly assessing its ability to meet multi-faceted objectives (representatively, space formation, stability, tractability, and symbiotic integration) for satisfactory GBR therapy. As the reviewer recognized, the medium-sized animal models have been generally favored in periodontal studies.^{7, 8} In particular, the canines offer excellent anatomical similarity to the human alveolar region; thereby, they allow us to simulate a periodontal lesion and analyze changes in the alveolar bone region.⁹ Hence, we believe that the canine models are a most valuable choice for studying

periodontal therapies and evaluating the efficacy of bone regeneration treatments. In the revised manuscript, we toned down the related descriptions toward discussing the observed data only in the canine model and not extending them to human perspectives.

Changes in manuscript reflecting Comment #3

(Page 20)

This canine premolar model offers excellent anatomical similarity to the human alveolar region.

(Page 22)

Although the SIOM's enhanced resistance towards dysbiosis was proved in vivo, there is a limited similarity (16.4%) of the bacterial taxa between canines and humans. Given this limitation, it would be premature to explore a mechanistic rationale with the currently available data.

Comment #4

How can you distinguish the effects from the membrane vs bone graft HAP/TCP material?

Authors' response to Comment #4

Thank you for the detailed and insightful comment. Our canine model aimed to simulate alveolar ridge regeneration therapy, which inevitably involves the application of a bone substitute supported by an occlusive membrane. To achieve this, we used HAP/TCP and employed a split-mouth design to deliver the same volume of graft to both sides. Therefore, the only variable was the type of membrane used, *i.e.*, CM vs. SIOM. The *in vivo* results demonstrate that both CM and SIOM showed an improvement in the new bone volume (NBV) compared to the blank group (spontaneous healing of uninflamed P3 socket). This result is consistent with the prior finding that a promising membrane therapy leads to increased performance of GBR therapy compared to spontaneous healing.¹⁰ Furthermore, the SIOM group exhibited significantly enhanced NBV values than the CM group. These results may be attributed to the SIOM-triggered local symbiotic microenvironment. In the revised manuscript, we clarified the experimental conditions of each group to avoid the misunderstanding of future readers.

Changes in manuscript reflecting Comment #4

(Page 24)

Following P2 extraction, the defective region was treated with a bone substitute in tandem with the SIOM or CM and subsequently closed with the inflamed gingival tissue (Fig. 6b). A P3 socket was prepared as a comparison reference and denoted as blank. Blank group means the untreated and uninflamed region under spontaneous healing without pin fixation.

Comment #5

How well would the intraoral scanner can establish for scanning dog teeth?

Normally the scanner is optimzised for human dentition.

Authors' response to Comment #5

Specially thank you for the helpful opinion, allowing us to clarify the methodology. We acknowledge that the primary application and design of intraoral scanners have been conducted for human dentition. However, we wish to persuade you that the working principle does not inherently limit itself to a human tissue form. The common working principles are Light Projection and Capture, as well as Distance to Object technologies.^{11, 12} Distance to Object technologies include triangulations, confocal, and active wave-front sampling. In this work, we used the Medit i500, which employs video-type scanning based on the triangulation technique.¹³ Furthermore, the image capture mechanism does not require the spray of pre-scan powder, allowing expeditious acquisition of 3D images from intraoral tissues in their natural state. Although the intraoral scanner's shape and size are typically adapted for the human oral cavity, it does not become a severe problem to apply in medium-sized animals such as mongrel dogs due to their wide mouth opening and anatomically longer jaws. Interestingly, recent studies have used similar scanning methodologies to assess intraoral tissues for analyzing soft-tissue grafts and implant impressions with scan bodies.^{14, 15}

Changes in manuscript reflecting Comment #5

(SI, Page 7)

Using an intraoral scanner (Medit i500), the digital impressions of the extraction site were taken up to 2 weeks after surgery. Because the employed intraoral scanner is operated through video-type scanning based on a triangulation technique, it allowed the expeditious acquisition of three-dimensional images from canine intraoral tissues.

Comment #6

I understand the membrane can be stretched quite a lot, so under the stretched condition, would the change of porosity or pore size affect the outcome?

Authors' response to Comment #6

We acknowledge the considerate comment allowing us to improve the manuscript quality. As the reviewer recognized, the satisfactory outcome of GBR therapy significantly depends on how much the occlusive membrane inhibits the fibroblast penetration into the periodontal alveolar defect. At the same time, the minimal pores should exist within the occlusive membrane for sufficient nutrient diffusion. In contrast, the micropore structure (100-300 μm pore size) could accumulate bacterial biofilm formation. Accordingly, the clinical standard PTFE membrane features submicron pores with $< 10 \mu\text{m}$ pore size.¹⁶ Considering the emerged opinion, we additionally conducted the mercury intrusion porosimeter (PM33GT, Quantachrome) experiments to figure out the pore sizes of the CM and SIOM. The clinical grade CM exhibited the desirable submicron size distribution (Figure R1a). Thanks to the complete coverage of HAPs, the SIOM featured a denser pore structure than CM (Figure R1b). In particular, a 10.6 μm -sized pore of CM was closed, and smaller pores (representatively, 0.034 μm size) were generated within SIOM. These results confirmed that the SIOM is advisable for successful GBR therapy in terms of structural occlusion.

Figure R1. Pore size distribution profiles of (a) CM and (b) SIOM.

As the reviewer kindly mentioned, the occlusive membrane should be pre-stretched before the implantation for tight closure during GBR therapy. The influence of pre-stretch in the pore size is highly complex and case-dependent. In other words, the 10% pre-stretch hardly correlates to a 10% increase in pore size. Meanwhile, recent works revealed that the pre-stretch might change the pore geometry by aligning the pores along the stretch direction.¹⁷ In this study, the CM and SIOM were also pre-stretched for *in vivo* experiments. Although the exact pore characteristics during the *in vivo* scenario were not determined, the outstanding GBR therapy with SIOM and moderate GBR outcome with CM better than the blank group imply that the variation in pore characteristics does not cause adverse effects such as fibroblast penetration.

Changes in manuscript reflecting Comment #6

(Page 9)

The mercury intrusion porosimeter experiments suggested that the SIOM featured a denser submicron pore structure than CM (Supplementary Fig. 5).

Comment #7

If the wound is being attacked by other bacteria, the symbiosis may be destroyed. How to proof your membrane is able to resist this kind of situation?

Authors' response to Comment #7

We appreciate the meaningful opinion regarding the performance of SIOM in an unfavorable microbial environment. As the reviewer mentioned, it was indeed a most necessary concern. Accordingly, we addressed this concern by designing both *in vitro* (*ex vivo*) and *in vivo* models in Figure 4-5 to involve a multispecies sample, not a single species as reported in various antibacterial studies.¹⁸ Moreover, the *in vivo* model in Figure 6 simulated an acute inflammation condition in which the wound is being exposed to the multispecies. Please note that we modified the term “*ex vivo*” to “*in vitro*” to clarify the human saliva experiment protocol of Figure 4.

Including a multispecies sample and an inflammatorily challenged condition allowed us to observe a selective microbial adhesion to the SIOM when the microbial interaction (representatively, cooperation, competition) prevails between the existing and/or intruding microorganisms. Notably, this microbiome community showed low to no pathogenic tendencies. We believe that these are the apparent verification of the effectiveness of the SIOM in unfavorable and complex microbial conditions (*e.g.*, being attacked by other bacteria).

Changes in manuscript reflecting Comment #7

(Page 26)

Notably, the SIOM exhibited GBR-therapy-favorable microbial interaction although it was exposed to both a multispecies sample and an inflammatorily challenged condition.

Comment #8

What is the degradation rate of your SIOM membrane? it looks like this is very stable...

Authors' response to Comment #8

We heartfully thank the reviewer for the meaningful comment. As the reviewer recognized, the structural stability of an occlusive membrane is a primary performance factor in achieving reliable GBR therapy.¹⁹ During several months of GBR therapy, the occlusive membrane must maintain its structure to provide sufficient time and space for closed alveolar bone augmentation. However, the premature resorption and degradation of the resorbable CM occurred due to the catabolic enzymes (representatively, collagenase type I) explosively secreted during the early regeneration phase.²⁰

To prove the outstanding enzymatic stability of SIOM, the CM and SIOM (1 x 1 cm²) were immersed in collagenase type I media (0.2 IU mL⁻¹) and incubated at 37 °C (Figure R2a). The CM degraded rapidly within 12 h, demonstrating its practical limitation with spacing durability. Remarkably, the residual SIOM was monitored even after 48 h of enzyme exposure. As shown in Figure R2b, the degradation rate was determined by assessing the residual membrane within 12 h. The CM exhibited a degradation rate of 0.035 h⁻¹; meanwhile, the SIOM presented a zero degradation rate. When the α -amylase media was used, the SIOM also showed a significant improvement in the enzymatic stability (Figure R2c). Moreover, the storage and loss moduli of the CM and SIOM were measured during 10 h of degradation (Figure R2d). The CM lost ~90% of its storage modulus after 4 h of degradation, implying that the

GBR zone dented rapidly. Meanwhile, the SIOM maintained its shear moduli even after 10 h of degradation. In conclusion, the SIOM exhibited outstanding enzymatic stability and long-lasting space maintenance properties. More detailed information can be found in Figure 3.

Figure R2. (a) Images of the CM and SIOM after being immersed in collagenase media. The orange arrows indicate the residual membranes. (b) Degradation rates of the CM and SIOM. (c) Amylase degradation profile of the CM and SIOM. (d) Rheological responses of the CM (left) and SIOM (right) during 10 h of collagenase degradation.

Changes in manuscript reflecting Comment #8

(Page 13)

The biomineralized NM and PM presented slightly decreased degradation rates: 0.035 h⁻¹ of CM, 0.025 h⁻¹ of NM, and 0.029 h⁻¹ of PM.

(Page 13)

The CM lost ~90% of its storage modulus after 4 h of degradation, implying that the GBR zone dented rapidly. On the other hand, the SIOM maintained its original properties (e.g., shear modulus and Ca/P ratio of 1.67) even after 10 h of degradation.

Reviewer #2

Reviewer's comments to author

Symbiotically Integrating Occlusive Membrane for Guided Regeneration of Inflammatory-Challenged Complex Tissue Defects, by Choi et al uses two assays to examining bacteria profiles on different membranes used to expedite healing and bone regeneration in the oral cavity. The ex vivo assay with human saliva is not really an ex vivo assay as samples are frozen in 30% glycerol prior to study, which kills many of the taxa. However, it is an interesting assay used to determine which oral bacteria will adhere and thrive on the surfaces of the various tested materials use to promote bone healing. This is done with the idea that some bacteria will interfere with the healing process. The second oral microbiome assay is done in vivo and relies on healing at a tooth extraction site in dogs with the different test materials used to promote healing. Bacteria samples are collected from crevicular fluid and from above CM and SIOM surface with paper points

Authors' response

The authors sincerely thank the reviewer for the significant and detail-oriented comments. Reflection on the comments has considerably helped towards bringing remarkable improvements in the manuscript. In particular, we endeavored to prevent the misunderstanding attributed to the ambiguous causative relation and focused on organisms identified at the species level. We wish the reviewer to positively review the remarkable manuscript changes as discussed below. For your information, we have rearranged and integrated your opinions to help the comprehension.

Comment #1

Ex vivo or in vitro assay:

Figure 4 would have more value if the taxa identification was done on the species level. For example, Porphyromonas gingivalis is a periodontal pathogen while other species of Porphyromonas in the oral cavity are not. Given the usage of human saliva for the ex vivo study I would suggest not use NCBI_16S_2021112 as the reference library for taxa identification and instead use a library restricted to human associated bacteria or that of the oral cavity. Or maybe sequence a different part of 16S rRNA gene.

Authors' response to Comment #1

Thank you for the beautiful suggestions. Acknowledging the reviewer's comment, we identified and analyzed sequences with the EZ-taxon database,²¹ which supports more than 98% Species level ID for body sub-site in the mouth such as saliva, mucosa, and plaque (https://www.ezbiocloud.net/resources/16s_download. Accessed 20th July 2023). In addition, we have entirely corrected the term “ex vivo assay” to “in vitro assay” for accuracy. In the revised manuscript, we have entirely updated the *in vitro* and *in vivo* results accordingly. Figure R3 presents the representative of the manuscript changes.

Figure R3. Revised Figure 4 with the species level resolution. (a) Taxonomic composition for the CM, ZM, and SIOM at the phylum-level in comparison to pooled saliva culture sequences. (b) Comparison of species-level relative abundance for taxa exhibiting the highest prevalence. Taxa representing less than 1% of total sequences were grouped as others. (c) Core microbiota Venn plot presenting species level shared and unique taxa by overlapping regions. (d) The dissimilarity percentage between the groups against the pooled saliva reference computed using SIMPER analysis. (e) The differences in abundances of taxonomic groups with greater than 1% contribution to dissimilarity to PC. The negative values in x axis indicate lower abundance in PC compared to CM/SIOM and vice-versa.

Changes in manuscript reflecting Comment #1

(Page 16)

More specific observations could also be made on resolving the differences at the species level resolution (Fig. 4e). The species with pathobiont nature, Fusobacterium periodonticum, Gemella haemolysans, Haemophilus parainfluenzae, Porphyromonas pasteri and Veillonella spp groups, remained undetected from the SIOM (Supplementary Fig. 12). Furthermore, SIOM exhibited a lower abundance of the bridging species, Veillonella, which leads to dysbiosis by acting as an accessory pathogen for keystone pathobionts. In particular, Veillonella parvula, Veillonella dispar, and Veillonella rogosae were found in higher concentrations in CM.

(Page 17)

Typically, the core microbiota is indicative of disease–disease relationship and it is varied by immediate environmental changes. The SIOM had a high co-occurrence representation of genera with PC, proving its potential for recovering microbial communities (Fig. 4f). In contrast, the ZM formed unique taxa and core microbiota distribution: Bacteroides vulgatus, Enterococcus faecium groups, and Romboutsia timonensis (Supplementary Table 3). In this context, the ZM was insufficient to induce a healthy microbiome although its zwitterionic groups nonspecifically prevented microbial growth up to 50% (Supplementary Fig. 14). Hence, ZMs without osteo-mimetic multiple nucleation biomineralization may be the tipping point for disturbed microbiomes.

(Page 17)

Fig. 4h draws the drawing comparison with the incubate of native pooled

salivary microbiota (PC), and Supplementary Fig. 15 shows the entire name of taxonomic groups. Regarding both CM and SIOM, the dissimilarity contribution was observed to be highest for Enterobacteriaceae and Bacillus cereus groups. However, V.dispar, P. melaninogenica, and F. periodonticum had ~3% dissimilarity contribution with higher abundance expressed at the CM interface. Furthermore, in specific periodontally related indicator species (Supplementary Note), the pathogenic Porphyromonas and Fusobacterium genera were observed in the PC and CM groups only.

(Page 18)

However, due to the sampling of saliva from a healthy cohort, it was limited in identifying the keystone species for periodontal pathogenesis using amplicon sequencing. Accordingly, we conducted Porphyromonas gingivalis and Staphylococcus aureus colony forming experiments to assess the specific microbial interaction (Supplementary Fig. 16). Remarkably, SIOM showed a 68% decreased P. gingivalis colony compared to the CM. Moreover, SIOM exhibited a 54% reduction in S. aureus colony formation. In particular, S.aureus is one of the pathogens with a favorable affinity towards titanium implants and found in the early microbiota after surgery. Considering that SIOM deployed near dental implants, its strong resistance against P. gingivalis and S. aureus further confirms its enhanced performance for satisfactory GBR outcome. Supplementary Fig. 17 shows the unsupervised pattern analysis with Spar CC method. Notably, the species in positive correlation to the identified pathobionts (P. pasteri, F. nuceatum, and F. periodonticum) had low relative abundance in SIOM. In other words, the SIOM exhibited a weaker correlation with the pathobionts. In conclusion, we confirm that the SIOM had an improved symbiosis-

inducing effect based on its favorable cellular and prokaryotic response.

(Page 21)

The alpha diversity metric at the sequence level evidenced a reduction in diversity for the SIOM compared to the CM, implying fewer unique operational taxonomic units in the SIOM (Supplementary Fig. 19). ANCOM-BC analysis suggested that both the CM and SIOM showed differentially abundant phylum-level expression when compared to the GCF (Supplementary Fig. 20). The beta-coefficient magnitude and its direction in Supplementary Table 4 indicate the degree of differential abundance with larger absolute values representing stronger differential abundance.

(Page 21)

Although CM and SIOM exhibited an increase in the relative abundance of the Fusobacteria content, SIOM presented a resistance towards Proteobacteria expression, with a progressive reduction up to the Sx+48 point (Fig. 5d). An increase of Proteobacteria expression in lesions has been observed when progressing to periodontitis in dogs. Accordingly, the monitored progressive reduction in the SIOM means its differential resistance to Proteobacteria. Furthermore, the SIOM featured an improved expression of the core and other microbiota with an overall 8.3% overlap with GCF (Fig. 5e). Moreover, the overlapping species were notably higher at 10.8% when resolving the SIOM's identified taxa at Sx+48 point (Supplementary Fig. 21). In contrast, the CM had an overall 3.6% overlap, which was particularly lower (2.2%) at the same time point. Thus, the SIOM was capable of resembling the native oral environment during the GBR therapy.

(SI, Page 5)

3.2. 16S rRNA sequencing assay

DNA sequencing was performed to identify the biofilm composition on specimens, as previously described. Briefly, the bacterial DNA was extracted from the samples using the DNeasy PowerSoil Kit following the manufacturer's instructions. Each sequenced sample was prepared according to the Illumina 16S rRNA sequencing library protocols. PicoGreen and VICTOR Nivo were employed to evaluate the quality and quantity of DNA. The 16S rRNA genes were then amplified using 16S V3-V4 primers, with an additional limited-cycle amplification step for the inclusion of multiplexing indices and Illumina sequencing adapters. Subsequently, the resulting products underwent normalization and pooling, and size confirmations, before being subjected to sequencing on the MiSeq™ platform.

For the amplicon sequencing, the Illumina Miseq Sequencing System from Illumina, USA was utilized. To ensure data integrity, a quality check on the raw reads, filtering out low-quality reads with a score of less than 25 was performed. Paired-end sequence data was merged, and the primers were trimmed. The unique 16S rRNA reads were isolated at similarity threshold of 97%, and allocated to taxonomy based on the EzBioCloud 16S rRNA database.

(SI, Page 7)

The collected specimens were filtered to remove bacteria and stored for sequence processing and analysis using the same methods as in the in vitro analysis. 16S rRNA Amplicon sequencing of V3 to V4 were analyzed using Bakt 341F-805R target region.

Comment #2

Furthermore, in specific periodontally related indicator species, the pathogenic Porphyromonas and Fusobacterium genera were observed in the PC and CM groups only (Figure 4h).

[Q1] This has more meaning when taxa are identified on species level.

[Q2] Need to identify these bacteria on species level as for example not all oral Porphyromonas that are in the periodontium or are pathogenic.

Authors' response to Comment #2

We sincerely thank the reviewer for awakening the importance of species-level identification. Accordingly, we re-performed the species-level evaluation of the *in vitro* and *in vivo* microbiome results. Reflecting on the reviewer's considerate opinion, we listed the specific genera of *Porphyromonas* and *Fusobacterium* evidenced by their clinical association with disease and studied their correlation with the other taxonomic groups. Specifically, we based on the identification and conducted the analysis of the SPAR CC correlation (unsupervised) pattern search (Figure R4).²² Notably, the species in positive correlation to the identified pathobionts (*Porphyromonas pasteri*, *Fusobacterium nucleatum*, and *Fusobacterium periodonticum*) had low relative abundance in the SIOM. Thus, we confirmed that the SIOM had a weaker microbial correlation with the pathobionts. Again, we heartfully thank the reviewer for enabling us to achieve this more profound and accurate insight.

Figure R4. The correlation patterns between *Porphyromonas pasteri*, *Fusobacterium nucleatum*, and *Fusobacterium periodonticum* were analyzed using the Spar CC method.²² This figure corresponds to Figure S17 in the revised manuscript.

Changes in manuscript reflecting Comment #2

(Page 16)

More specific observations could also be made on resolving the differences at the species level resolution (Fig. 4e). The species with pathobiont nature, *Fusobacterium periodonticum*, *Gemella haemolysans*, *Haemophilus parainfluenzae*, *Porphyromonas pasteri* and *Veillonella* spp groups, remained undetected from the SIOM (Supplementary Fig. 12). Furthermore, SIOM exhibited a lower abundance of the bridging species, *Veillonella*, which leads to dysbiosis by acting as an accessory pathogen for keystone pathobionts. In particular, *Veillonella parvula*, *Veillonella dispar*, and *Veillonella rogosae* were found in higher concentrations in CM.

(Page 18)

Supplementary Fig. 17 shows the unsupervised pattern analysis with Spar CC method. Notably, the species in positive correlation to the identified pathobionts (*P. pasteri*, *F. nucleatum* and *F. periodonticum*) had low relative abundance in SIOM. In

other words, the SIOM exhibited a weaker correlation with the pathobionts. In conclusion, we confirm that the SIOM had an improved symbiosis-inducing effect based on its favorable cellular and prokaryotic response.

Comment #3

[Figure S11. Core abundance index (CAI)] CAI provides a useful metric for assessing the health of microbial ecology by focusing on the most important species and their abundance. It represents the averaged abundances of core species identified (>80% prevalence). A high CAI indicates a well-functioning microenvironment with adequate populations of key species, whereas a low CAI may indicate stress or disturbance.

It is hard to know the point of this figure. It seems to indicate what the other figures show, that with usage of SIOM or ZM the average abundance of the core taxa is much higher. Isn't this just another way of saying that with the usage of these 2 materials there is much less variety of taxa present, so the ones present are at higher relative abundance. Contrary to what is suggested in the figure legend above loss of diversity is often associated with disease, though in periodontal disease that may be controversial. It is interesting that the different materials used have different taxa adhering, but it is premature to claim that that is because fewer pathogens are present, though the authors can certainly speculate about it. Without species level identification they have not even shown association with periopathogens.

Authors' response to Comment #3

We absolutely sympathize with the reviewer's concern that the original investigation on the CAI profile (*i.e.*, addressing the periodontal pathogens and associated diseases) was overexpressed. After studying the species level correlation and considering the reviewer's entire comments involving from #1 to #10, we determined to remove the CAI-related Figure S11. By revealing the species level

features of the microbiome community, we found that the species with a pathobiont nature, representatively, *Fusobacterium periodonticum*, *Gemella haemolysans*, *Haemophilus parainfluenzae*, *Porphyromonas pasteri* and *Veillonella* spp groups, were not detected from the SIOM (Figure R5). Furthermore, the SIOM exhibited a lower abundance of the bridging species, *Veillonella*, which leads to dysbiosis by acting as an accessory pathogen for keystone pathobionts.^{23, 24} In particular, *Veillonella parvula*, *Veillonella dispar*, and *Veillonella rogosae* were found in higher concentrations in CM. Based on these findings, we clarified that the microbiome community at the SIOM prefers commensal oral microbiota with a reduced abundance of pathobionts. At the same time, we refined several statements that the SIOM could alleviate periodontal diseases because we have learned that the original manuscript somewhat jumped over the undiscussed causative relation between microbiota and disease.

Figure R5. Species level comparison of CM and SIOM. Log abundance distribution pattern observed for the *Fusobacterium*, *Prevotella*, *Streptococcus*, and *Veillonella*

genera. PC represents the microbiota expression from pooled salivary incubate without interaction in the absence of occlusive membranes. This figure corresponds to Figure S12 in the revised manuscript.

Box R2. Interaction between specific periodontal pathogens and the SIOM

Contemplating the reviewer's comment, we reached the curiosity about SIOM's correlation with known periodontal pathogens, not with the potential pathobionts above. However, we encountered a limitation because the conducted *in vitro* multi-species biofilm assay condition was not favorable for pathogen colonization. In detail, we utilized saliva samples obtained from healthy human adults with no signs of periodontal ailments or recent history of such conditions. Moreover, the incubation environment we used resembled the native oral cavity rather than a predominantly anaerobic sub-gingival habitat. Given these factors, the task of identifying species level differences regarding the keystone species for periodontal pathogenesis was a tough challenge.

Instead, we additionally conducted a *Porphyromonas gingivalis* and *Staphylococcus aureus* colony forming experiments to figure out the specific association with perio-pathogens (Figure R6). In particular, we conducted the colony forming unit (CFU) analysis of single *Porphyromonas gingivalis* species *in vitro* (Figure R6a). The observations agreed with the multispecies results, showing remarked reductions in colonization of *Porphyromonas gingivalis* (~68%). Among broad applications of occlusive membranes, a common use-case is associated with the placement of titanium-based implants. However, the implant surfaces are vexed with *Staphylococcus aureus* secondary colonization²⁵. Taking it account, we also

performed CFU evaluation for *Staphylococcus aureus* and observed a ~54% reduction in the colony forming units (Figure R6b).

Figure R6. CFU of (a) *Porphyromonas gingivalis*, (b) *Staphylococcus aureus* co-cultured with CM and SIOM. Figure R6 corresponds to Figure S16 in the revised manuscript.

Changes in manuscript reflecting Comment #3

(Page 16)

As shown in the relative abundance at genus level (Supplementary Fig. 11), three periodontal pathobionts serving as taxonomic biomarkers, namely, *Prevotella*, *Alloprevotella*, and *Actinomyces*, did not exhibit any abundance at SIOM. More specific observations could also be made on resolving the differences at the species level resolution (Fig. 4e). The species with pathobiont nature, *Fusobacterium periodonticum*, *Gemella haemolysans*, *Haemophilus parainfluenzae*, *Porphyromonas pasteri* and *Veillonella* spp groups, remained undetected from the SIOM (Supplementary Fig. 12). Furthermore, SIOM exhibited a lower abundance of the bridging species, *Veillonella*, which leads to dysbiosis by acting as an accessory pathogen for keystone pathobionts. In particular, *Veillonella parvula*, *Veillonella dispar*,

and Veillonella rogosae were found in higher concentrations in CM. This result is associated with the actual lack of gamma diversity as the Hill number increased (Supplementary Fig. 13). Typically, a shifted microbiome is described as a niche-specific species turnover (gain or loss). Accordingly, the near total lack of pathobionts (especially *Bacteroidetes* and *Actinobacteria*) indicates that the microbiome community at the SIOM prefers commensal oral microbionts.

(Page 17)

Accordingly, the near total lack of pathobionts (especially *Bacteroidetes* and *Actinobacteria*) indicates that the microbiome community at the SIOM prefers commensal oral microbionts.

(Page 17)

Fig. 4h draws the drawing comparison with the incubate of native pooled salivary microbiota (PC), and Supplementary Fig. 15 shows the entire name of taxonomic groups. Regarding both CM and SIOM, the dissimilarity contribution was observed to be highest for *Enterobacteriaceae* and *Bacillus cereus* groups. However, *V. dispar*, *P. melaninogenica*, and *F. periodonticum* had ~3% dissimilarity contribution with higher abundance expressed at the CM interface. Furthermore, in specific periodontally related indicator species (Supplementary Note), the pathogenic *Porphyromonas* and *Fusobacterium* genera were observed in the PC and CM groups only.

(Page 18)

However, due to the sampling of saliva from a healthy cohort, it was limited in identifying the keystone species for periodontal pathogenesis using amplicon sequencing. Accordingly, we conducted *Porphyromonas gingivalis* and *Staphylococcus aureus* colony forming experiments to assess the specific microbial interaction (Supplementary Fig. 16). Remarkably, SIOM showed a 68% decreased *P. gingivalis* colony compared to the CM. Moreover, SIOM exhibited a 54% reduction in *S. aureus* colony formation. In particular, *S. aureus* is one of the pathogens with a favorable affinity towards titanium implants and found in the early microbiota after surgery. Considering that SIOM deployed near dental implants, its strong resistance against *P. gingivalis* and *S. aureus* further confirms its enhanced performance for satisfactory GBR outcome. Supplementary Fig. 17 shows the unsupervised pattern analysis with Spar CC method. Notably, the species in positive correlation to the identified pathobionts (*P. pasteri*, *F. nucleatum*, and *F. periodonticum*) had low relative abundance in SIOM. In other words, the SIOM exhibited a weaker correlation with the pathobionts. In conclusion, we confirm that the SIOM had an improved symbiosis-inducing effect based on its favorable cellular and prokaryotic response.

Comment #4

The authors need to discuss more whether this reduction in diversity is seen in the *in vivo* studies using the canine model. Figure 5 d is only on the phylum level. Is the alpha diversity lower on the DNA sequence level for the SIOM versus CM membranes as the *ex vivo* study indicated?

Authors' response to Comment #4

We especially thank the reviewer for the helpful comment and apologize for the insufficient details of the *in vivo* experiments in the original description. During the revision process, the low diversity observed in the *in vitro* microbiota analysis was monitored as well in the open healing *in vivo* model (Figure R7). Specifically, considering the alpha diversity metric for the *in vivo* results, the SIOM exhibited a lower diversity at the sequence level variants, particularly compared to the CM. As the reviewer suggested, the Chao1 is the estimate of the true species richness, and the abundance coverage estimator (ACE) accounts for the observed and undetected species. Both Chao1 and ACE presented a lower metric for SIOM, implying that there are fewer unique operational taxonomic units.

Figure R7. SIOM's reduced alpha diversity at the level of amplicon sequence variants. Figure R7 corresponds to Figure S19 in the revised manuscript.

Changes in manuscript reflecting Comment #4

(Page 21)

The alpha diversity metric at the sequence level evidenced a reduction in diversity for the SIOM compared to the CM, implying fewer unique operational taxonomic units in the SIOM (Supplementary Fig. 19).

(SI, Page 28)

The observed diversity provides a direct count of species in a sample. The observed diversity provides a direct count of species in a sample. Chao1 and Abundance-based Coverage Estimator (ACE) diversity indices estimate the total species richness, considering both observed and unobserved species.

Comment #5

There is nothing wrong with the authors suggestion that the SIOM membrane works well due to lack of inflammation associated bacteria. However, with the large decreases in taxa number, approx. 7x, it is hard to identify specific bacteria taxa responsible for differences if they indeed are causative.

Authors' response to Comment #5

Firstly, we appreciate your affirmation of the outstanding advances of the SIOM. We completely acknowledge that the present study's findings are not definitive to state a causative relation with periodontal inflammation diseases. Thanks to your entire comments, we were able to learn a more logical methodology in the microbiome investigation; that is, we should be especially cautious about deducing a specific reason within a complexly entangled microbiome community. During the revision process, we trimmed concerned overexpressing descriptions not to jump over the causality and focused on the definitive relative comparison within the microbiome community variation.

Box R3. Discussion about reduced inflammatory expression with the SIOM

Regarding that you have mentioned the keyword inflammation, we wish to address further how the SIOM resulted in the reduced expression of inflammatory markers, in this Box R3. While the reported microbial resistance technologies can exhibit bactericidal and/or bacteriostatic outcomes under an apparent action mechanism, they would be limited to fulfill the essential requirements (especially

symbiotic integration) of successful GBR therapy. This limitation becomes particularly significant in the inflammatory-challenged condition on which Figure 6 was based. The newly studied results, colony formation units in Figure S14 and alpha diversity in Figure S19, support that the SIOM exhibited excellent resistance to microbial adhesion. We believe that this microbial resistance of the SIOM might positively affect the minimal expression of MPO, CD86, and CD20. In contrast, as proved in Figure 4, an individual antifouling performance of the ZM was not capable of achieving symbiosis. Therefore, we wish to persuade you that the microbial resistance of the SIOM is a part of symbiosis integration; thereby, the observed pro-inflammatory cytokine expression data is completely attributed to the SIOM itself. Unfortunately, the microbiota sampling from the closed model in Figure 6 could not be conducted due to methodological design. We strongly believe that further exploratory analysis of the *in vivo* microbiota changes will assist the understanding of the same. Again, regardless of the discussion in Box R3, we toned down the entire manuscript to just deliver the facts supported by reasonable results and causality.

Changes in manuscript reflecting Comment #5

(Page 18)

*However, due to the sampling of saliva from a healthy cohort, it was limited in identifying the keystone species for periodontal pathogenesis using amplicon sequencing. Accordingly, we conducted *Porphyromonas gingivalis* and *Staphylococcus aureus* colony forming experiments to assess the specific microbial interaction (Supplementary Fig. 16). Remarkably, SIOM showed a 68% decreased *P. gingivalis* colony compared to the CM. Moreover, SIOM exhibited a 54% reduction in*

S. aureus colony formation. In particular, *S.aureus* is one of the pathogens with a favorable affinity towards titanium implants and found in the early microbiota after surgery. Considering that SIOM deployed near dental implants, its strong resistance against *P. gingivalis* and *S. aureus* further confirms its enhanced performance for satisfactory GBR outcome. Supplementary Fig. 17 shows the unsupervised pattern analysis with Spar CC method. Notably, the species in positive correlation to the identified pathobionts (*P. pasteri*, *F. nuceatum*, and *F. periodonticum*) had low relative abundance in SIOM. In other words, the SIOM exhibited a weaker correlation with the pathobionts. In conclusion, we confirm that the SIOM had an improved symbiosis-inducing effect based on its favorable cellular and prokaryotic response.

(Page 20)

A balanced and diverse oral environment, i.e., symbiosis, promotes physiological healing, while the absence of symbiosis precipitates an environment that interrupts the healing process.

Comment #6

[Table S4] Please have columns with relative levels of taxa on CM and SIOM and maybe the FDR value for each comparison. Additionally, define Beta-coefficient(w). There needs to be more of an effort to make the results understandable.

Authors' response to Comment #6

The authors agree entirely with this considerate comment. According to the feedback, we have updated the ANCOM-BC-related Supporting Information to include the relative abundance information and the interpretation key. Here, the beta coefficient represents the effect size indicating the change in the relative abundance of taxa between the groups being compared.

Changes in manuscript reflecting Comment #6

(Page 21)

ANCOM-BC analysis suggested that both the CM and SIOM showed differentially abundant phylum-level expression when compared to the GCF (Supplementary Fig. 20). The beta-coefficient magnitude and its direction in Supplementary Table 4 indicate the degree of differential abundance with larger absolute values representing stronger differential abundance.

(SI, Page 32)

Supplementary Table 4. FDR (false discovery rate) bias-corrected analysis of differentially abundant taxa with analysis of compositions of microbiomes with bias correction (ANCOM-BC)

Phyla	Beta-coefficient (W)		Standard error		Differentially abundant		Relative abundance		FDR value (q)	
	CM	SIOM	CM	SIOM	CM	SIOM	CM	SIOM	CM	SIOM
Tenericutes	-2.08695	-0.86879184	0.530866	0.63809377	FALSE	FALSE	3.33	1.33	5.74E-02	4.49E-01
Spirochaetes	-2.090111	-3.75868513	1.02842	0.76510948	FALSE	TRUE	197.33	44.00	5.74E-02	5.86E-04
Proteobacteria	1.160967	0.28546156	0.695246	0.47726797	FALSE	FALSE	1374.00	852.00	3.44E-01	7.75E-01
Fusobacteria	0.220584	3.07034844	0.834404	0.32180151	FALSE	TRUE	9678.67	10601.00	8.25E-01	4.28E-03
Bacteroidetes	0.500062	1.2329256	0.374642	0.31016438	FALSE	FALSE	10836.67	10717.33	6.81E-01	3.05E-01
Actinobacteria	-1.041663	-1.02959918	0.588714	0.56279316	FALSE	FALSE	108.33	132.67	3.79E-01	3.86E-01
Verrucomicrobia	4.633183	3.707471	0.21341	0.27674468	TRUE	TRUE	0.00	0.00	1.68E-05	5.86E-04
Synergistetes	-9.247514	-9.62984912	0	0	TRUE	TRUE	0.00	0.33	0.00E+00	0.00E+00
Planctomycetes	4.692319	4.20788622	0.231824	0.26213618	TRUE	TRUE	0.33	0.33	1.68E-05	1.20E-04
Other*	-2.463879	-1.71939438	0.658528	0.8269246	TRUE	FALSE	14.33	176.33	2.75E-02	1.33E-01
Firmicutes	3.963607	5.21980338	0.283852	0.22013747	TRUE	TRUE	11653.00	10296.00	2.58E-04	1.25E-06
Cyanobacteria	3.278702	2.91790054	0	0	TRUE	TRUE	1.67	2.67	0.00E+00	0.00E+00
Chloroflexi	-0.47846	-0.47881451	0	0	TRUE	TRUE	0.00	0.00	0.00E+00	0.00E+00
Actinobacteria	3.215367	3.33564011	0.448713	0.32717334	TRUE	TRUE	108.33	132.67	3.79E-01	3.86E-01

#ANCOMBC2 identifies taxa that exhibit significant differences in abundance between groups. The beta coefficient (W) is a measure of the effect size representing the change in the relative abundance of a taxon between the groups being compared. It provides an estimate of the magnitude and direction of change in abundance associated with the group. The magnitude and direction of the beta coefficient indicate the degree of differential abundance. Larger absolute values of W suggest stronger differential abundance, while values closer to zero indicate less pronounced differences in abundance between groups. FDR(q) values present the adjusted p values by Benjamini & Hochberg correction of false discovery rate.

Comment #7

Q1. [Line 317] *Dysbiosis scores presented using out-of-bag (OOB) probability scores and median community-level variations (CLVs). Presumably a healthy microbiome is based on what is present prior to surgery. The reference for dysbiosis, Wei, S., Bahl, M.I., Baunwall, S.M.D., Hvas, C.L. & Licht, T.R. Determining gut microbial dysbiosis: a review of applied indexes for assessment of intestinal microbiota imbalances. Appl. Environ. Microbiol. 87, e00395-00321 (2021) does not explain CLV.*

Q2. *Presumably what the authors are saying is that during healing with SIOM the variation of the bacteria taxa seen are most similar to the healthy state. It is a little puzzling why Fusobacteria earlier stated as being linked to disease would increase with SIOM. More effort needs to be made to explain differences in in vitro (ex vivo) and in vivo results in regard to microbiome.*

Authors' response to Comment #7

We apologize for the ambiguity of the stated reference and the lack of clarity with the results. Here is the answer to the first question Q1. The description for median CLV refers to the "Category 3.1 Neighborhood classification method," as described by *Wei et al.*²⁶ The method was used for the calculation of dysbiosis score where the selection measure was median variation.²⁷ These calculations were implemented with dysbiosisR package (<https://github.com/microsud/dysbiosisR>).

As mentioned in the second concern Q2, the authors truly acknowledge that the *Fusobacteria*-related descriptions could lead to misunderstanding. Thanks to the reviewer's entire comments, we found that this concern also originates from the hazy

explanation based on the ambiguous causative relation. Therefore, we toned down the related discussion based on the relative comparison within the overall microbiome variation trend (Figure R8a). In detail, we focused on that relative abundance analysis revealed an increase in the *Fusobacteria* content in both CM and SIOM, while only SIOM presented a resistance towards *Proteobacteria* expression, with a progressive reduction up to the Sx+48 point. Specifically, *Proteobacteria* expression has been observed to increase in lesions when progressing to periodontitis in dogs.²⁸ Furthermore, we figured out the apparent effectiveness of SIOM by studying the core microbiota distribution at the species level resolution (Figure R8b). This native oral microbiome-resembled core microbiota distribution was available to be discovered because we were able to broaden it up to the species level after reflecting on the reviewer’s wonderful comment. Consequently, we have modified the corresponding sections of the manuscript as suggested below.

Figure R8. Revised version of Figure 5 to discuss the relative comparison within the microbiome community variation. (a) Taxonomic relative abundance at the phylum level presented across the sampling timeline. (b) Core microbiota distribution between the GCF, CM, and SIOM, observed at the species level resolution.

Changes in manuscript reflecting Comment #7

(Page 21)

Although CM and SIOM exhibited an increase in the relative abundance of the Fusobacteria content, SIOM presented a resistance towards Proteobacteria expression, with a progressive reduction up to the Sx+48 point (Fig. 5d). An increase of Proteobacteria expression in lesions has been observed when progressing to periodontitis in dogs. Accordingly, the monitored progressive reduction in the SIOM means its differential resistance to Proteobacteria. Furthermore, the SIOM featured an improved expression of the core and other microbiota with an overall 8.3% overlap with GCF (Fig. 5e). Moreover, the overlapping species were notably higher at 10.8% when resolving the SIOM's identified taxa at Sx+48 point (Supplementary Fig. 21). In contrast, the CM had an overall 3.6% overlap, which was particularly lower (2.2%) at the same time point. Thus, the SIOM was capable of resembling the native oral environment during the GBR therapy.

(Page 21)

Two statistical indexes were used to quantify the symbiosis-inducing performance of the SIOM: OOB predicted probability and CLVs (Figure 5f). OOB and CLV (neighborhood classification method by community) are scoring indexes for disturbed symbiosis, *i.e.*, dysbiosis.

Comment #8

[Line 286] to study the microbiome development at the SIOM, we used metagenomic sequencing of co-cultured salivary extracts from six healthy adults (Figure 4c)

Please use the term 16S rRNA sequencing. Metagenomic sequencing is usually reserved for shotgun sequencing of multiple organisms.

Authors' response to Comment #8

Thank you for the helpful suggestion. To improve the accuracy of the methodology description, the manuscript has been revised to reflect the reviewer's feedback, as shown below.

Changes in manuscript reflecting Comment #8

(Page 16)

To study the microbiome development at the SIOM, we used 16S rRNA sequencing of co-cultured salivary extracts from six healthy adults (Fig. 4c).

(SI, Page 5)

3.2. 16S rRNA sequencing assay

DNA sequencing was performed to identify the biofilm composition on specimens, as previously described.²⁹ Briefly, the bacterial DNA was extracted from the samples using the DNeasy PowerSoil Kit following the manufacturer's instructions. Each sequenced sample was prepared according to the Illumina 16S rRNA

sequencing library protocols.

(SI, Page 7)

This protocol was done to ensure repeatability and quality control in the sampling protocol. The collected specimens were filtered to remove bacteria and stored for sequence processing and analysis using the same methods as in the *in vitro* analysis. 16S rRNA Amplicon sequencing of V3 to V4 were analyzed using Bakt 341F-805R target region.

Comment #9

[Line 278] Thus, a settled microbiome community with a unilaterally low F/B scan (impedes- delete). (May Impede-add) normal healing through a chronic expression of pro-inflammatory cytokine. Speculation is stated as fact.

Authors' response to Comment #9

Thank you for the considerate comment. We have amended the manuscript following the reviewer's suggestion.

Changes in manuscript reflecting Comment #9

(Page 16)

Thus, a settled microbiome community with an unilaterally low F/B scan may impede normal healing through a chronic expression of pro-inflammatory cytokines.

Comment #10

[Line 360] An established microbial symbiosis results in an expeditious regeneration response. As proof, we observed an improvement in the open healing rate of the gingival tissue during GBR therapy with the SIOM (Figure 5g).

The result supports the conclusion but is by no means proof of it. The claim needs to be toned down.

Authors' response to Comment #10

We truly acknowledge your constructive criticism of our work. We gained a lot of help through suggestions, which have contributed to the notable improvement of the manuscript. We have clarified the implications of data by suggesting the physio-functions of HAPs and the reference addressing the importance of the microbiome in the regeneration process.³⁰ Please review the toned-down descriptions in the manuscript below.

Changes in manuscript reflecting Comment #10

(Page 22)

Fig. 5g showed the rapid gingival tissue regeneration under the SIOM-triggered well-established microbial symbiosis. Notably, the SIOM presented a marked reduction in the early surgical wound size within Sx+72. Furthermore, it achieved near complete healing in less than one week of GBR therapy (Sx+168). At Sx+168, the SIOM and CM exhibited significant differences ($p = 0.002$). When the CM was deployed, the healing efficiency of the SIOM at Sx+168 took twice as long (Sx+336)

to achieve. Although HAP has outstanding physio-functions in promoting hard tissue regeneration and cellular differentiation, its relevance to soft tissue regeneration is rarely reported. On the other hand, the importance of microbiota composition in the systemic regeneration process has been gradually revealed. In this context, Fig. 5g implies that the SIOM-based symbiosis induction significantly helps an expeditious healing response during GBR therapy.

Reviewer #3

Reviewer's comments to author

The authors present a significant advancement in the field of oral tissue engineering with their development of a symbiotically integrating occlusive membrane (SIOM). This innovative membrane is designed to promote the regeneration of a biologically durable tooth-like environment, particularly at the interface of closed healing and open healing regions. The study investigates the potential of SIOM biomineralization and enzymatic durability, as well as its ability to induce symbiosis, ex vivo and in vivo. An important aspect that warrants further exploration is the influence and interaction of the oral microbiome with the engineered membranes. Understanding this interaction is crucial for comprehensive analysis and should be considered in future research. The work is interesting and novel, offering promising considerations for more realistic oral soft or hard tissue engineering approaches. The authors provide comprehensive information regarding the synthesis and characterizations of the SIOM membrane, demonstrating a strong foundation for their findings. However, to meet the criteria for publication in Nature Communications, some clarifications and justifications are necessary. These improvements will further enhance the quality and impact of the research:

The authors truly appreciate the thorough review and significant comments. Below are our detailed efforts to completely respond to your feedback. If the reviewer finds any further concerns, the authors hopefully wish to address them. For your information, we have integrated some opinions to help the comprehension.

Comment #1

The authors used the rheological assessment to study the SIOM mechanical properties after biomineralization using indicators of storage modulus and $\tan(\delta)$. The $\tan(\delta)$ shows the ratio of lost energy to stored energy during cyclic deformation. How can this information demonstrate the mechanical properties of hard and soft tissue? The proper informative explanation is missing. Please provide more discussion.

Authors' response to Comment #1

The authors appreciate the thorough review. We improved the readability by reflecting on the reviewer's comment. As the reviewer mentioned, the loss or damping factor ($\tan\delta$) represents the energy dissipation property.³¹ Accordingly, the characteristic viscoelasticity of biological tissues results in the diverse values of their loss factors.^{32, 33} Moreover, the significance of the developed material's viscoelasticity in biomedical applications (e.g., tissue organization,³⁴ bioelectronics³⁵) was established. Recent works (representatively, *Science* (2022), *Adv. Funct. Mater* (2021)^{36, 37}) employed the loss factor to express the viscoelasticity of the developed materials. In this work, we investigated the viscoelasticity of the symbiotically integrating occlusive membrane (SIOM) through the storage modulus vs. loss factor profile. In detail, we referred to Figure 2c of *Science* (2022) when preparing the storage modulus vs. loss factor plot (Figure R9a).³⁶

Thanks to the reviewer's comment, we found that the original Figure 2h should be revised to deliver more accurate information. *Nature* (2020) revealed that the loss factors of most biological materials involving extracellular matrix, soft tissues, and hard

tissues are in the range of 0.1-0.2 (Figure R9b).³² Through revising Figure 2h as shown in Figure R9c, we concluded that the SIOM was highly similar to the human soft and hard tissues in terms of viscoelasticity. This mechanical feature implies that the SIOM is particularly desirable for GBR therapy, considering the occlusive membrane will be deployed between soft and hard tissues.

Figure R9. (a) Storage modulus vs. loss factor plot extracted from Figure 2c of *Science* (2022).³⁶ (b) Linear relationship between loss modulus and storage modulus of biological materials. Blue, yellow, and black mean the extracellular matrix, soft tissues, and hard tissues, respectively. This graph is obtained from Figure 2 of *Nature* (2020).³² (c) Revised Figure 2c presenting the accurate loss factor range of biological tissues involving soft and hard tissues.

Changes in manuscript reflecting Comment #1

(Page 9)

When a developed material is applied to a specific tissue, it should exhibit similar viscoelasticity to surrounding tissues for a satisfactory operation. The viscoelasticity of SIOM was studied through the storage modulus vs. loss factor profile (Fig. 2h). The loss factors of most biological materials involving extracellular matrix,

soft tissues, and hard tissues are in the range of 0.1-0.2. Remarkably, the viscoelasticity of the SIOM was comparable to the biological tissues. Therefore, SIOM is physically beneficial when deployed at the interface of the gingival gum and the alveolar bone defect.

Comment #2

SIOM membrane is the biomineralized zwitterionic membrane (ZM). The authors mentioned that the mechanical properties of a fully covered HAP (mineralized) membrane (which is a ZM membrane) decreased, but SIOM showed higher mechanical properties (Figure S5). How SIOM presented higher mechanical properties than ZM, while SIOM is a biomineralized form of ZM? This feature requires proper justification.

Authors' response to Comment #2

We apologize that the original explanation of Figure S5 was ambiguous. The zwitterionic membrane (ZM) was prepared by the zwitteration of the collagen membrane (CM). The SIOM was obtained after the multiple nucleation biomineralization of the ZM. Zwitteration was performed by immersing CM in the precursor solution and subsequent 365 nm UV irradiation. In particular, we employed the dual-radical strategy to achieve the zwitterated CM, not acquire the zwitterionic hydrogel independent of CM.³⁸ To this end, two photo-initiators, benzophenone and 2-hydroxy-4'-(2-hydroxyethoxy)-2-methylpropiophenone, were used. If the free radicals are captured by the charged side group of Arg or Lys, the peptide bond could be dissociated.³⁹ This oxidative stress might cause the reduction of mechanical properties in ZM compared to CM. Because hydroxyapatite (HAP) is a highly strong biomineral and superior in energy dissipation,⁴⁰ the introduction of HAP leads to a significant improvement in mechanical properties. For instance, when HAP was introduced to the acrylamide hydrogel, it enhanced the ultimate compression stress

up to 2 times and ultimate tensile stress up to 3 times.^{41, 42} This HAP filler effect was also observed in the SIOM, resulting in an increase in mechanical properties compared to the ZM. Remarkably, the ultimate stress and Young's modulus of the SIOM (3.32 MPa and 7.65 MPa, respectively) were recovered to the level of clinical grade CM. We demonstrated this detailed information in the revised manuscript.

Changes in manuscript reflecting Comment #2

(SI, Page 15)

If the free radicals are captured by the charged side group of Arg or Lys, the peptide bond could be dissociated. Due to the harsh radical condition, the mechanical property is reduced in the ZM compared to the CM. Because the strong HAP is outstanding in energy dissipation, the introduction of HAP always leads to a significant improvement in mechanical properties. For instance, when HAP was introduced to the acrylamide hydrogel, it enhanced the ultimate compression stress up to 2 times and ultimate tensile stress up to 3 times. In terms of the SIOM, the biomineralized HAP compensates for the decreased mechanical property of the ZM. Remarkably, the SIOM shows the comparable property with clinical grade CM. In detail, the ultimate stress and Young's modulus of the SIOM are 3.32 MPa and 7.65 MPa, respectively. A uniform tensile deformation of 5.0 mm min^{-1} was applied to the beam-type samples (4.0 cm^2) through a universal testing machine (Model 3366, Instron, USA).

Comment #3

The oral environment harbors a vast array of bacterial species, numbering over 700. Figure 4 in the study presents data regarding the specific genus under investigation. It would greatly enhance the manuscript if the authors could provide a concise explanation for the rationale behind studying these particular genera. Such information would offer valuable insights into the selection process and help readers understand the significance of focusing on these specific bacterial groups within the context of the research.

Comment #6

Furthermore, it would greatly enhance the manuscript if the authors could provide clarification and a comprehensive list of the healthy and diseased oral microbiome (specifically ones that were studied in this project). This clarification would facilitate comparisons between the outcomes of the study and existing literature, providing valuable insights into the potential applications and limitations of the SIOM membrane in various oral health contexts.

Authors' response to Comment #3 and #6

The authors sincerely thank you for the helpful suggestions. By performing the 16S rRNA amplicon sequencing, we identified the taxonomic level distribution of *in vitro* and *in vivo* (canine model) studies. For data reliability, we have suggested a list of the identified core microbiota in Table S3. Remarkably, the observed results imply that the structural modification of the CM into SIOM alters the taxa-adhering behavior.

For your information, we modified the term “*ex vivo*” to “*in vitro*” for clarifying the human saliva experiment protocol of Figure 4.

The clusters of bacterial species have been identified to correlate with each other and present in higher abundance in specific clinical cases (e.g. progression from gingivitis to periodontitis). Therefore, our rational selection for the particular genera regarded the genera of which their association with periodontal diseases was clinically revealed. To this end, we employed the clustering of bacteria species from plaque reported by *Socransky, Haffajee, Cugini, Smith and Kent*, which is widely accepted in periodontal practice.⁴³ This classification is suggested as Supplementary Note in the revised manuscript. However, because the complex role of most organisms is niche-specific, a definitive classification into healthy or disease-associated microbiomes might not be applicable for generalization. In other words, a clear distinction was cautious in this study, considering the complex microbial interactions that impact the taxa associations.

Instead, reflecting on the reviewer’s considerate feedback, we listed the taxa evidenced by their clinical association with disease and studied their correlation with the other taxonomic groups. Specifically, we based on the species level identification of the most prevalent taxa and conducted the analysis of the SPAR CC correlation (unsupervised) pattern search (Figure R10).²² These results gave a detailed review of the taxonomic units of interest in the present study. Notably, the species in positive correlation to the identified diseased-related taxonomic units (*Porphyromonas pasteri*, *Fusobacterium nucleatum*, *Veillonella atypica*, and *Fusobacterium periodonticum*) had low relative abundance in SIOM. Thus, we confirmed that the SIOM had a weaker microbial correlation with the pathobionts.

Figure R10. The correlation patterns between *Porphyromonas pasteri*, *Fusobacterium nucleatum*, *Fusobacterium periodonticum* and *Veillonella atypica* were analyzed using the Spar CC method. This statistical approach helps assess the associations between different taxa in microbiome datasets, particularly when dealing with compositional data and was performed using web-based platform.⁴⁴ This figure is suggested as Figure S17 in the revised manuscript.

Changes in manuscript reflecting Comment #3 and #6

(Page 18)

Furthermore, in specific periodontally related indicator species (Supplementary Note), the pathogenic Porphyromonas and Fusobacterium genera were observed in the PC and CM groups only.

(Page 18)

*Supplementary Fig. 17 shows the unsupervised pattern analysis with Spar CC method. Notably, the species in positive correlation to the identified pathobionts (*P. pasteri*, *F. nucleatum* and *F. periodonticum*) had low relative abundance in SIOM. In other words, the SIOM exhibited a weaker correlation with the pathobionts.*

(Attached file)

Supplementary Table S3. Core taxonomic groups identified from downstream analysis of in vitro co-culture of the human salivary microbiome.

(SI, Page 9)

The Socransky classification is employed to categorize periopathogens, which are bacteria associated with periodontal diseases. This classification scheme, based on the pioneering work of Socransky et. al using checkerboard DNA-DNA hybridization, aids in understanding the complexity of the oral microbiome and its impact on oral

health. The classification groups periopathogens into six complexes (clusters) based on their association with periodontal disease severity:

- Orange Complex (initial stages)

Fusobacterium nucleatum subspecies, *Prevotella intermedia*, *Prevotella nigrescens*, *Peptostreptococcus micros*, *Campylobacter rectus*, *Campylobacter showae*, *Campylobacter gracilis*, *Eubacterium nodatum*, and *Streptococcus constellatus*

- Green Complex (early to moderate periodontitis)

Capnocytophaga species, *Campylobacter concisus*, *Eikenella corrodens*, and *Actinobacillus actinomyceteticus*

- Yellow Complex (moderate disease)

Streptococcus mitis, *Streptococcus sanguis* and *Streptococcus oralis*

- Purple Complex (advanced periodontitis)

Veillonella parvula and *Actinotnyces odontolyticus*

- Red Complex (severe disease)

Porphyromonas gingivalis, *Bacteroides forsythus* and *Treponema denticola*)

(SI, Page 26)

The correlation patterns between *Porphyromonas pasteri*, *Fusobacterium nucleatum*, and *Fusobacterium periodonticum* were analyzed using the Spar CC method. This statistical approach helps assess the associations between different taxa in microbiome datasets, particularly when dealing with compositional data and was performed using web-based platform.

Comment #4

Authors studied osteogenic differentiation using qPCR. However, the duration of the study is missing. By specifying the duration, the authors can effectively demonstrate the extent to which the SIOM promotes osteogenic differentiation and highlight its potential for long-term or short-term tissue regeneration.

Authors' response to Comment #4

We appreciate the thorough comment. In this study, the human bone marrow stem cells (BMSCs) and CM or SIOM were cultured in growth media and osteogenic differentiation media for 2 weeks. Subsequently, a quantitative real-time polymerase chain reaction (qPCR) was performed to observe the expression degree of osteogenic biomarkers. Interestingly, OPN expression increased by 267% ($p = 0.0317$) in BMSCs cultured in the growth media with the SIOM, thereby indicating the increased density of osteogenic differentiation. According to BMSC tests conducted under differentiation media conditions, the SIOM exhibited significantly enhanced expression of differentiation markers: a 127% increase in RUNX2, a 118% increase in OCN, a 1,065% increase in OPN ($p = 0.000012$), and a 2,884% increase in BMP2 ($p = 0.000249$). These significantly enhanced expressions of OPN and BMP2 suggested that the SIOM promotes bone healing up to the bone matrix deposition stage during 2 weeks.⁴⁵ Accordingly, the SIOM is desirable for the GBR therapy of several months by promptly regenerating the alveolar bone and consequently preventing the penetration of fibroblast and epithelial cells.

Changes in manuscript reflecting Comment #4

(Page 15)

BMSCs and CM or SIOM were cultured in growth media and osteogenic differentiation media for 2 weeks. Subsequently, a quantitative real-time polymerase chain reaction was performed to study the expression degree of osteogenic biomarkers.

(Page 15)

In particular, the significantly enhanced expressions of OPN and BMP2 suggested that SIOM promotes bone healing up to the bone tissue formation stage during 2 weeks. Therefore, SIOM is desirable for the GBR therapy of several months requiring the expeditious augmentation of the alveolar bone.

(SI, Page 4)

BMSCs were cultured in growth media and osteogenic differentiation media for 2 weeks.

Comment #5

Regarding the discussion of Figure 4f, it will improve the article significantly if the authors correlate data with native oral microbiome diversity and frequency.

Authors' response to Comment #5

The authors truly thank the reviewer for the valuable suggestions. As the reviewer mentioned, we provided an *in vitro* analysis of unperturbed pooled saliva obtained from healthy adult individuals (referred to as PC) in the original Figure 4. In other words, we wish to persuade you that the comparison with a typical human oral microbiome was performed in this work. Acknowledging the reviewer's comment, we identified and analyzed sequences with the EZ-taxon database,²¹ which supports more than 98% species level ID for human body sub-site in the mouth such as saliva, mucosa, and plaque. In other words, we correlated the data obtained by dealing with the native oral microbiome sequences, reflecting on the reviewer's feedback. Subsequently, we elaborated on the distinct factors as the species level resolution as depicted in Figure R11.

Figure R11a suggests the taxonomic composition of the CM, ZM, and SIOM at the phylum level in comparison to pooled saliva culture sequences. Figure R11b shows the comparison of species-level relative abundance for taxa exhibiting the highest prevalence. Here, taxa representing less than 1% of total sequences were grouped as others. Figure R11c is the core microbiota Venn plot presenting species level shared and unique taxa by overlapping regions. Figure R11d presents the dissimilarity percentage between the groups against the pooled saliva reference

computed using SIMPER analysis. Figure R11e demonstrates the differences in abundances of taxonomic groups with greater than 1% contribution to dissimilarity to PC. The negative values in x axis indicate lower abundance in PC compared to CM/SIOM and vice-versa.

Figure R11. Revised version of Figure 4. This figure is prepared by correlating with the native oral microbiome database (EZ-taxon database).

Changes in manuscript reflecting Comment #5

(Page 16)

More specific observations could also be made on resolving the differences at the species level resolution (Fig. 4e). The species with pathobiont nature,

Fusobacterim periodonticum, *Gemella haemolysans*, *Haemophilus parainfluenzae*, *Porphyromonas pasteri* and *Veillonella* sps groups, remained undetected from the SIOM (Supplementary Fig. 12). Furthermore, SIOM exhibited a lower abundance of the bridging species, *Veillonella*, which leads to dysbiosis by acting as an accessory pathogen for keystone pathobionts. In particular, *Veillonella parvula*, *Veillonella dispar*, and *Veillonella rogosae* were found in higher concentrations in CM.

(Page 17)

Typically, the core microbiota is indicative of disease–disease relationship and it is varied by immediate environmental changes. The SIOM had a high co-occurrence representation of genera with PC, proving its potential for recovering microbial communities (Fig. 4f). In contrast, the ZM formed unique taxa and core microbiota distribution: *Bacteroides vulgatus*, *Enterococcus faecium* groups, and *Romboutsia timonensis* (Supplementary Table 3). In this context, the ZM was insufficient to induce a healthy microbiome although its zwitterionic groups nonspecifically prevented microbial growth up to 50% (Supplementary Fig. 14). Hence, ZMs without osteo-mimetic multiple nucleation biomineralization may be the tipping point for disturbed microbiomes.

(Page 17)

Fig. 4h draws the drawing comparison with the incubate of native pooled salivary microbiota (PC), and Supplementary Fig. 15 shows the entire name of taxonomic groups. Regarding both CM and SIOM, the dissimilarity contribution was observed to be highest for *Enterobacteriaceae* and *Bacillus cereus* groups. However,

V. dispar, *P. melaninogenica*, and *F. periodonticum* had ~3% dissimilarity contribution with higher abundance expressed at the CM interface. Furthermore, in specific periodontally related indicator species (Supplementary Note), the pathogenic *Porphyromonas* and *Fusobacterium* genera were observed in the PC and CM groups only.

(Page 18)

However, due to the sampling of saliva from a healthy cohort, it was limited in identifying the keystone species for periodontal pathogenesis using amplicon sequencing. Accordingly, we conducted *Porphyromonas gingivalis* and *Staphylococcus aureus* colony forming experiments to assess the specific microbial interaction (Supplementary Fig. 16). Remarkably, SIOM showed a 68% decreased *P. gingivalis* colony compared to the CM. Moreover, SIOM exhibited a 54% reduction in *S. aureus* colony formation. In particular, *S. aureus* is one of the pathogens with a favorable affinity towards titanium implants and found in the early microbiota after surgery. Considering that SIOM deployed near dental implants, its strong resistance against *P. gingivalis* and *S. aureus* further confirms its enhanced performance for satisfactory GBR outcome. Supplementary Fig. 17 shows the unsupervised pattern analysis with Spar CC method. Notably, the species in positive correlation to the identified pathobionts (*P. pasteri*, *F. nucleatum*, and *F. periodonticum*) had low relative abundance in SIOM. In other words, the SIOM exhibited a weaker correlation with the pathobionts. In conclusion, we confirm that the SIOM had an improved symbiosis-inducing effect based on its favorable cellular and prokaryotic response.

Comment #7

[Q1] According to Figure 5, it is evident that the CM membranes exhibit a core-microbiota distribution that is similar to the gingival crevicular fluid (GCF). This similarity is indeed favorable as it indicates a resemblance to the native oral environment.

[Q2] However, it is important for the authors to clearly justify why the SIOM membrane is superior to the CM membranes, considering that the primary aim was to mimic the native tissue environment. It would be also valuable if the authors could discuss any functional or structural features of the SIOM that directly facilitate the establishment and maintenance of a healthy oral microbiome. Exploring how the SIOM promotes symbiosis and influences the microbial community in a beneficial manner would further substantiate the claim that it provides a more realistic and advantageous condition for oral tissue engineering.

Authors' response to Comment #7

The authors completely agree with the reviewer's opinion. Reflecting on the reviewer's first comment Q1, we re-evaluated the core microbiota at the species level resolution as suggested in Figure R12. Considering the core microbiota distribution of SIOM at Sx+48 point, the identified taxa presented an overall 10.8% overlap with GCF. In contrast, the CM exhibited only a 2.2% overlap with GCF. Thus, the SIOM exhibited an improved expression of the core and other microbiota, sufficiently resembling the native oral environment. Again, we heartfully thank the reviewer for enabling us to achieve this more profound and accurate insight.

Figure R12. Core-microbiota distribution between the GCF, CM, and SIOM, observed at different time points. This figure corresponds to Figure S21 in the revised manuscript.

As the reviewer mentioned in Q2, the primary idea for SIOM-based symbiosis induction was the establishment of a structurally and biologically tooth enamel-like environment during the GBR therapy. To reach this concept, we have focused on our tooth, which endogenously retains symbiosis regardless of the diverse microbes that exist in the oral cavity. In particular, the tooth's outer layer, *i.e.*, enamel, significantly contributes to the maintenance of a healthy oral microbiome. The species with pathogenic potentials, representatively, *Streptococcus* and *Porphyromonas*, also dwell in the homeostatic biofilms on exposed enamel surfaces.⁴⁶ If these pathogenic members reach harmful colonization, they demineralize the enamel, construct the anaerobic cavity, aggregate into acidic caries, and finally damage the entire oral microbiome.^{47, 48} While the pathogenic colonization is determined by several factors (*e.g.*, sugar-rich diet, salivary condition, oral hygiene),⁴⁹ the enamel physically endures the demineralization and inhibits the cavitation thanks to its structural feature: stable and dense HAPs. In other words, stable and dense HAPs are the robust barrier to retaining symbiosis. Motivated by this intriguing property, we designed SIOM to be

entirely analogous with enamel, from the biomineralization mechanism to the structural feature of HAPs.

The key obstacle in realizing the considerably tooth enamel-like SIOM was the multiple nucleations of the enamel through the controlled ion adsorption with biological ion receptors (*e.g.*, OPN, DMP1).^{50, 51} We successfully emulated the tooth enamel biomineralization mechanism (*i.e.*, multiple nucleation biomineralization) by designing the ZM, of which the zwitterionic groups functioned as selectively ion adsorbing nucleation sites, similar to OPN and DMP1. Specifically, the positive moiety of the ZM ($-N^+R^4-$) preferentially interacted with the PO_4^{3-} ions, and the negative moiety ($-SO_3^-$) was bound to Ca^{2+} . Remarkably, after the ZM-assisted multiple nucleation biomineralization, the obtained SIOM was fully covered with HAPs, proving that it is highly similar to enamel's dense HAP structure.⁵² As proved by *in vitro*, *ex vivo*, and *in vivo* experiments, a considerably enamel-like environment was constructed when SIOM was employed for the GBR therapy, leading to the symbiotic oral microbiome community and satisfactory outcome of GBR.

Because the natural biomineralization mechanism of enamel was thoroughly implemented in this work,⁵³ we wish to persuade that it is distinctive compared to the prior works developing various biomineralization technologies, for instance, hydrothermal,⁵⁴ enzyme,⁵⁵ electrochemical,⁵⁶ and photo-mediated⁵⁷ strategies.

Changes in manuscript reflecting Comment #7

(Page 5)

Our tooth endogenously retains symbiosis regardless of the diverse microbes

in the oral cavity. In particular, the tooth's outer layer, i.e., enamel, significantly contributes to the maintenance of a healthy oral microbiome. The pathobionts, representatively, Streptococcus and Porphyromonas, also dwell in the homeostatic biofilms on exposed tooth surfaces. If these keystone pathogens form harmful colonization, they demineralize the tooth enamel, construct the anaerobic cavity, aggregate into acidic caries, and finally damage the entire oral microbiome. While pathogenic colonization is controlled by several factors (e.g., sugar-rich diet, salivary condition, oral hygiene), the tooth enamel physically prevents demineralization and cavitation through its robust barrier structure: durable and dense hydroxyapatites (HAPs).

(Page 5)

Inspired by this naturally maintained symbiosis, we developed an occlusive membrane that is entirely analogous to tooth enamel, from the HAP growth mechanism to dense barrier HAPs. To this end, we established enamel-mimetic multiple nucleation biomineralization of HAP in a resorbable CM. The obtained membrane is called a symbiotically integrating occlusive membrane (SIOM). When the SIOM was employed for the GBR therapy, it might construct a considerably enamel-like environment at the interface of closed healing and open healing regions, inducing symbiosis.

(Page 9)

In other words, the application of SIOM means an establishment of a durable and tooth enamel-like ordered structural environment during GBR therapy.

(Page 21)

Furthermore, the SIOM featured an improved expression of the core and other microbiota with an overall 8.3% overlap with GCF (Fig. 5e). Moreover, the overlapping species were notably higher at 10.8% when resolving the SIOM's identified taxa at Sx+48 point (Supplementary Fig. 21). In contrast, the CM had an overall 3.6% overlap, which was particularly lower (2.2%) at the same time point. Thus, the SIOM was capable of resembling the native oral environment during the GBR therapy.

Comment #8

In in vivo canine closed healing study, pins were used to fix CM and SIOM membranes. Pin fixation may cause an additional inflammatory response which is one of the limitations required to be discussed. Pin fixation may also explain the high MPO, CD86, and CD20 expression of the CM membrane in comparison with the Blank (control) group. In addition, the experimental results suggest that the pin fixation method used in the study did not have an adverse impact on the properties of SIOM. SIOM demonstrated similar behavior to the Blank group in terms of the expression of pro-inflammatory cytokines. Providing a strong, detailed and well-supported explanation for the superiority of the SIOM over the CM membranes will address this point of concern.

Authors' response to Comment #8

In alignment with the reviewer's comment, we truly acknowledge that pin fixation can cause an inflammatory response. Currently, the fixation of the occlusive membranes serves the purpose of preventing membrane dislocation and minimizing the transfer of micro-movements to the augmented site.⁵⁸ In other words, this fixation is inevitable in safeguarding the bone augmentation site and improving GBR therapy outcome.⁵⁹ Accordingly, during *in vivo* experiments, we adopted the pin fixation-based membrane stabilization method in both CM and SIOM groups. To address the pin-triggered inflammatory reaction issue, we designed the blank group not to be subjected to the pin fixation step. In other words, the type of occlusive membrane was the principal variable rather than the presence of a pin. According to the experimental

results, (1) the similar pro-inflammatory cytokines expressions between the blank and the SIOM groups and (2) significantly increased cytokines expression only in the CM group indicate the pin hardly presented an adverse effect in this study and prove the superior GBR therapy outcome with the SIOM.

Moreover, regarding the inflammatory-challenged condition of our closed healing model, the peri-pin infection could be a potential source of bacterial infection. The newly updated results in Figure R13-14 support that the SIOM exhibited an excellent resistance to microbial adhesion. We believe that this microbial resistance of the SIOM might positively affect the minimal expression of MPO, CD86, and CD20. In contrast, as proved in Figure 4, an individual antifouling performance of the ZM was not capable of achieving symbiosis. Therefore, we wish to argue that the microbial resistance of the SIOM is a part of the symbiosis induction property; thereby, the observed pro-inflammatory cytokines expression data is completely attributed to the SIOM itself. Unfortunately, the microbiota sampling from the closed model was not able to be conducted in this study. We strongly believe that further exploratory analysis of the *in vivo* microbiota changes during closed healing will suggest a deeper insight.

Figure R13. Reduction in the alpha diversity at the level of amplicon sequence

variants *in vivo*. Chao1 and ACE diversity indices estimate the total species richness, considering both observed and unobserved species. Figure R13 corresponds to Figure S19 in the revised manuscript.

Figure R14. Colony formation unit (CFU) of (a) *Porphyromonas gingivalis*, (b) *Staphylococcus aureus* co-cultured with CM and SIOM. Figure R14 corresponds to Figure S16 in the revised manuscript.

Changes in manuscript reflecting Comment #8

(Page 18)

Accordingly, we conducted a *Porphyromonas gingivalis* and *Staphylococcus aureus* colony forming experiments to assess the specific microbial interaction (Supplementary Fig. 16). Remarkably, SIOM showed a 68% decreased *P. gingivalis* colony compared to the CM. Moreover, SIOM exhibited a 54% reduction in *S. aureus* colony formation. In particular, *S. aureus* is one of the pathogen with favorable affinity towards titanium implants and found in the early microbiota after surgery. Considering that SIOM deployed near dental implants, its strong resistance against *P. gingivalis*

and S. aureus further confirms its enhanced performance for satisfactory GBR outcome.

(Page 21)

The alpha diversity metric at the sequence level evidenced a reduction in diversity for the SIOM compared to the CM, implying fewer unique operational taxonomic units in the SIOM (Supplementary Fig. 19).

(Page 24)

Following P2 extraction, the defective region was treated with a bone substitute in tandem with the SIOM or CM and subsequently closed with the inflamed gingival tissue (Fig. 6b). A P3 socket was prepared as a comparison reference and denoted as blank. Blank group means the untreated and uninflamed region under spontaneous healing without the pin fixation.

Comment #9

Line 261-264 needs adequate citations. Line 360 requires citations.

Authors' response to Comment #9

We acknowledge that the proper references were missed in the mentioned sections and apologize for the mistakes. Regarding the original lines 261-264, we referred to *Nat. Rev. Mol. Cell. Biol* (2020) and *Nat. Rev. Rheumatol* (2023).^{60, 61} Reflecting on your comment #4, we toned down the line 261-264 to demonstrate the positive correlation between the biomarker expression level and osteogenic potential, not suggesting their time sequences. Although line 360 referred to *Front. Cell. Dev. Biol* (2022),³⁰ it was trimmed to focus on the observed experimental data, as suggested below.

Changes in manuscript reflecting Comment #9

(Page 15)

The above biomarker results indicate that the osteo-mimetic SIOM mediated an excellent stimulation for osteo-differentiation.

(Page 22)

On the other hand, the importance of microbiota composition in the systemic regeneration process has been gradually revealed. In this context, Fig. 5g implies that the SIOM-based symbiosis induction significantly helps an expeditious healing response during GBR therapy.

Reviewer #4

Reviewer's comments to author

I co-reviewed this manuscript with one of the reviewers who provided the listed reports as part of the Nature Communications initiative to facilitate training in peer review and appropriate recognition for co-reviewers

The authors sincerely appreciate the considerable contribution in reviewing our work. The emerging opinions were meaningful and helped us greatly improve the manuscript. We heartfully wish this response letter could relieve your entire feedback completely.

References

1. Botelho, J. *et al.* Economic burden of periodontitis in the United States and Europe: An updated estimation. *J. Periodontol.* **93**, 373-379 (2022).
2. Li, S. *et al.* Hard tissue stability after guided bone regeneration: a comparison between digital titanium mesh and resorbable membrane. *Int. J. Oral Sci.* **13**, 37 (2021).
3. Aprile, P., Letourneur, D. & Simon-Yarza, T. Membranes for guided bone regeneration: a road from bench to bedside. *Adv. Healthc. Mater.* **9**, 2000707 (2020).
4. Chen, K. *et al.* Recent Advances in the Development of Magnesium-Based Alloy Guided Bone Regeneration (GBR) Membrane. *Metals* **12**, 2074 (2022).
5. Zhang, T., Zhang, T. & Cai, X. The application of a newly designed L-shaped titanium mesh for GBR with simultaneous implant placement in the esthetic zone: a retrospective case series study. *Clin. Implant. Dent. Relat. Res.* **21**, 862-872 (2019).
6. Li, B. *et al.* Oral bacteria colonize and compete with gut microbiota in gnotobiotic mice. *Int. J. Oral Sci.* **11** (2019).
7. Albuquerque, C. *et al.* Canine periodontitis: the dog as an important model for periodontal studies. *Vet. J.* **191**, 299-305 (2012).
8. Weinberg, M.A. & Bral, M. Laboratory animal models in periodontology. *J Clin Periodontol* **26**, 335-340 (1999).
9. Wikesjö, U.M. *et al.* The critical-size supraalveolar peri-implant defect model: characteristics and use. *J. Clin. Periodontol.* **33**, 846-854 (2006).
10. Troiano, G. *et al.* Combination of bone graft and resorbable membrane for

- alveolar ridge preservation: A systematic review, meta-analysis, and trial sequential analysis. *J. Periodontol.* **89**, 46-57 (2018).
11. Logozzo, S., Zanetti, E.M., Franceschini, G., Kilpelä, A. & Mäkynen, A. Recent advances in dental optics – Part I: 3D intraoral scanners for restorative dentistry. *Opt. Lasers Eng.* **54**, 203-221 (2014).
 12. Richert, R. *et al.* Intraoral Scanner Technologies: A Review to Make a Successful Impression. *J. Healthc. Eng.* **2017**, 1-9 (2017).
 13. Khaled, M., Sabet, A., Ebeid, K. & Salah, T. Effect of Different Preparation Depths for an Inlay-Retained Fixed Partial Denture on the Accuracy of Different Intraoral Scanners: An In Vitro Study. *J. Prosthodont.* **31**, 601-605 (2022).
 14. Lee, K.S. *et al.* Dimensional ridge changes in conjunction with four implant timing protocols and two types of soft tissue grafts: A pilot pre-clinical study. *J. Clin. Periodontol.* **49**, 401-411 (2022).
 15. Huang, R. *et al.* Improved accuracy of digital implant impressions with newly designed scan bodies: an in vivo evaluation in beagle dogs. *BMC Oral Health* **21** (2021).
 16. Ronda, M., Rebaudi, A., Torelli, L. & Stacchi, C. Expanded vs. dense polytetrafluoroethylene membranes in vertical ridge augmentation around dental implants: a prospective randomized controlled clinical trial. *Clin. Oral. Implants. Res.* **25**, 859-866 (2014).
 17. Lin, S., Liu, J., Liu, X. & Zhao, X. Muscle-like fatigue-resistant hydrogels by mechanical training. *Proc. Natl. Acad. Sci. U.S.A* **116**, 10244-10249 (2019).
 18. Kim, M. *et al.* Antimicrobial PEGtides: A modular poly (ethylene glycol)-based peptidomimetic approach to combat bacteria. *ACS Nano* **15**, 9143-9153 (2021).

19. Bottino, M.C. *et al.* Recent advances in the development of GTR/GBR membranes for periodontal regeneration—A materials perspective. *Dent. Mater.* **28**, 703-721 (2012).
20. Kania, A. & Klein, R. Mechanisms of ephrin–Eph signalling in development, physiology and disease. *Nat. Rev. Mol. Cell Biol.* **17**, 240-256 (2016).
21. Yoon, S.-H. *et al.* Introducing EzBioCloud: a taxonomically united database of 16S rRNA gene sequences and whole-genome assemblies. *Int. J. Syst. Evol. Microbiol.* **67**, 1613-1617 (2017).
22. Friedman, J. & Alm, E.J. Inferring Correlation Networks from Genomic Survey Data. *PLoS Comput. Biol.* **8**, e1002687 (2012).
23. Hoare, A. *et al.* A cross-species interaction with a symbiotic commensal enables cell-density-dependent growth and in vivo virulence of an oral pathogen. *ISME J.* **15**, 1490-1504 (2021).
24. Zhou, P., Manoil, D., Belibasakis, G.N. & Kotsakis, G.A. Veillonellae: Beyond Bridging Species in Oral Biofilm Ecology. *Front. Oral Health* **2**, 774115 (2021).
25. Furst, M.M., Salvi, G.E., Lang, N.P. & Persson, G.R. Bacterial colonization immediately after installation on oral titanium implants. *Clin. Oral Implants Res.* **18**, 501-508 (2007).
26. Wei, S., Bahl, M.I., Baunwall, S.M.D., Hvas, C.L. & Licht, T.R. Determining Gut Microbial Dysbiosis: a Review of Applied Indexes for Assessment of Intestinal Microbiota Imbalances. *Appl. Environ. Microbiol.* **87** (2021).
27. Lloyd-Price, J. *et al.* Multi-omics of the gut microbial ecosystem in inflammatory bowel diseases. *Nature* **569**, 655-662 (2019).
28. Wallis, C. *et al.* Subgingival microbiota of dogs with healthy gingiva or early

- periodontal disease from different geographical locations. *BMC Vet. Res.* **17** (2021).
29. Mangal, U. *et al.* Polybetaine-enhanced hybrid ionomer cement shows improved total biological effect with bacterial resistance and cellular stimulation. *Biomater. Sci.* (2023).
 30. Díaz-Díaz, L.M., Rodríguez-Villafañe, A. & García-Arrarás, J.E. The role of the microbiota in regeneration-associated processes. *Front. Cell Dev. Biol.* **9**, 768783 (2022).
 31. Huang, J. *et al.* Ultrahigh energy-dissipation elastomers by precisely tailoring the relaxation of confined polymer fluids. *Nat. Commun.* **12**, 3610 (2021).
 32. Chaudhuri, O., Cooper-White, J., Janmey, P.A., Mooney, D.J. & Shenoy, V.B. Effects of extracellular matrix viscoelasticity on cellular behaviour. *Nature* **584**, 535-546 (2020).
 33. Burla, F., Mulla, Y., Vos, B.E., Aufderhorst-Roberts, A. & Koenderink, G.H. From mechanical resilience to active material properties in biopolymer networks. *Nature Reviews Physics* **1**, 249-263 (2019).
 34. Elosegui-Artola, A. *et al.* Matrix viscoelasticity controls spatiotemporal tissue organization. *Nat. Mater.* **22**, 117-127 (2023).
 35. Tringides, C.M. *et al.* Viscoelastic surface electrode arrays to interface with viscoelastic tissues. *Nat. Nanotechnol.* **16**, 1019-1029 (2021).
 36. Zhao, H. *et al.* Multiscale engineered artificial tooth enamel. *Science* **375**, 551-556 (2022).
 37. Ma, Y. *et al.* Viscoelastic cell microenvironment: hydrogel-based strategy for recapitulating dynamic ECM mechanics. *Adv. Funct. Mater.* **31**, 2100848 (2021).

38. Yu, Y. *et al.* Multifunctional “hydrogel skins” on diverse polymers with arbitrary shapes. *Adv. Mater.* **31**, 1807101 (2019).
39. Turecek, F. N C α Bond Dissociation Energies and Kinetics in Amide and Peptide Radicals. Is the Dissociation a Non-ergodic Process? *J. Am. Chem. Soc.* **125**, 5954-5963 (2003).
40. Jiang, W. *et al.* The effects of hydroxyapatite coatings on stress distribution near the dental implant–bone interface. *Appl. Surf. Sci.* **255**, 273-275 (2008).
41. Nonoyama, T. *et al.* Double-network hydrogels strongly bondable to bones by spontaneous osteogenesis penetration. *Adv. Mater.* **28**, 6740-6745 (2016).
42. Li, Z. *et al.* A tough hydrogel–hydroxyapatite bone-like composite fabricated in situ by the electrophoresis approach. *J. Mater. Chem. B* **1**, 1755-1764 (2013).
43. Socransky, S.S., Haffajee, A.D., Cugini, M.A., Smith, C. & Kent, R.L., Jr. Microbial complexes in subgingival plaque. *J. Clin. Periodontol.* **25**, 134-144 (1998).
44. Chong, J., Liu, P., Zhou, G. & Xia, J. Using MicrobiomeAnalyst for comprehensive statistical, functional, and meta-analysis of microbiome data. *Nat. Protoc.* **15**, 799-821 (2020).
45. Albeshri, S. *et al.* Biomarkers as Independent Predictors of Bone Regeneration around Biomaterials: A Systematic Review of Literature. *J. Contemp. Dent. Pract* **19**, 605-618 (2018).
46. Tuganbaev, T., Yoshida, K. & Honda, K. The effects of oral microbiota on health. *Science* **376**, 934-936 (2022).
47. Lamont, R.J., Koo, H. & Hajishengallis, G. The oral microbiota: dynamic communities and host interactions. *Nat. Rev. Microbiol.* **16**, 745-759 (2018).

48. Kolenbrander, P.E., Palmer Jr, R.J., Periasamy, S. & Jakubovics, N.S. Oral multispecies biofilm development and the key role of cell–cell distance. *Nat. Rev. Microbiol.* **8**, 471-480 (2010).
49. Bowen, W.H., Burne, R.A., Wu, H. & Koo, H. Oral biofilms: pathogens, matrix, and polymicrobial interactions in microenvironments. *Trends Microbiol.* **26**, 229-242 (2018).
50. Grohe, B. *et al.* Control of calcium oxalate crystal growth by face-specific adsorption of an osteopontin phosphopeptide. *J. Am. Chem. Soc.* **129**, 14946-14951 (2007).
51. George, A. & Veis, A. Phosphorylated proteins and control over apatite nucleation, crystal growth, and inhibition. *Chem. Rev.* **108**, 4670-4693 (2008).
52. Yeom, B. *et al.* Abiotic tooth enamel. *Nature* **543**, 95-98 (2017).
53. De Yoreo, J.J. *et al.* Crystallization by particle attachment in synthetic, biogenic, and geologic environments. *Science* **349**, aaa6760 (2015).
54. Xu, B. *et al.* A mineralized high strength and tough hydrogel for skull bone regeneration. *Adv. Funct. Mater.* **27**, 1604327 (2017).
55. Rauner, N., Meuris, M., Zoric, M. & Tiller, J.C. Enzymatic mineralization generates ultrastiff and tough hydrogels with tunable mechanics. *Nature* **543**, 407-410 (2017).
56. He, C., Xiao, G., Jin, X., Sun, C. & Ma, P.X. Electrodeposition on nanofibrous polymer scaffolds: rapid mineralization, tunable calcium phosphate composition and topography. *Adv. Funct. Mater.* **20**, 3568-3576 (2010).
57. Wei, H. *et al.* Visible-light-mediated nano-biomineralization of customizable tough hydrogels for biomimetic tissue engineering. *ACS Nano* **16**, 4734-4745

(2022).

58. Kacarevic, Z.P. *et al.* Biodegradable magnesium fixation screw for barrier membranes used in guided bone regeneration. *Bioact. Mater.* **14**, 15-30 (2022).
59. Mir-Mari, J., Wui, H., Jung, R.E., Hammerle, C.H. & Benic, G.I. Influence of blinded wound closure on the volume stability of different GBR materials: an in vitro cone-beam computed tomographic examination. *Clin. Oral Implants Res.* **27**, 258-265 (2016).
60. Salhotra, A., Shah, H.N., Levi, B. & Longaker, M.T. Mechanisms of bone development and repair. *Nat. Rev. Mol. Cell Biol.* **21**, 696-711 (2020).
61. Duda, G.N. *et al.* The decisive early phase of bone regeneration. *Nature Reviews Rheumatology* **19**, 78-95 (2023).

REVIEWER COMMENTS

Reviewer #1 (Remarks to the Author):

I have reviewed the response and the revised article, and they look acceptable for me, and can be accepted now.

Reviewer #2 (Remarks to the Author):

Symbiotically Integrating Occlusive Membrane for Guided Regeneration of Inflammatory-Challenged Complex Tissue Defect, by Choi et al. has been improved with the addition of experiments and more detail. The authors look at membrane effects on healing and associated oral microbiome. I appreciate the authors' efforts in this resubmission. I just have a few minor requests remaining on this work that in my view is near ready for acceptance.

It is good that they were able to show that in the canine model there is also a reduction in richness of amplicon sequence read variants as one might expect. It is good that they use ANCOM-BC a program for looking at differential sequence abundance that as that corrects well technical differences in samples.

That authors have demonstrated large difference in microbiome associated with CM, ZM and SIOM. They have improved their case that SIOM retards *P. gingivalis* and other pathogens *S. aureus*.

"For the amplicon sequencing, the Illumina Miseq Sequencing System from Illumina, 100 USA was utilized. To ensure data integrity, a quality check on the raw reads, filtering out low-quality reads with a score of less than 25 was performed. Paired-end sequence data was merged and the primers were trimmed. The unique 16S rRNA reads were isolated at similarity threshold of 97%, and allocated to taxonomy based on the EzBioCloud 16S rRNA database"

In the supplement there needs to be at least a reference to the method used to merge paired sequence and the identification of similar sequence reads (DADA2?) unless it is elsewhere in the paper and I missed it.

I do not understand supplemental table 4. If FDR is >0.1 for a comparison that is usually not considered significantly differentially abundant. Like in the case of Actinobacteria. If another system is used to define statistical significance it needs to be even more clearly expressed.

However, the oral microbe-related material technologies are still bound to bacterial-free functionalities, for instance, antifouling 49, 74 and bactericidal effects.

This is just a suggestion but may want to change that to: However, the oral microbe-related material technologies still are focused on total prevention of bacteria accumulation, for instance, antifouling 49, 74 and bactericidal effects 75, 76, 77.

The above may or may not be clearer.

Reviewer #3 (Remarks to the Author):

Dear authors,

Thank you for addressing my comments. I have no additional queries and the manuscript has greatly improved.

Reviewer #4 (Remarks to the Author):

I co-reviewed the revised manuscript with one of the reviewers who provided the listed reports as part of the Nature Communications initiative to facilitate training in peer review and appropriate recognition for co-reviewers.

Response Letter to Reviewer #2

Manuscript ID: NCOMMS-23-24435A

Title: Symbiotically Integrating Occlusive Membrane for Guided Regeneration of Inflammatory-Challenged Complex Tissue Defects

Author: Woojin Choi† , Utkarsh Mangal† , Ji Yeong Na, Ji-Yeong Kim, Taesuk Jun, Ju Won Jung, Moonhyun Choi, Sungwon Jung, Milae Lee, Jin-Young Park, Du Yeol Ryu, Jin-Man Kim, Jae-Sung Kwon, Won-Gun Koh, Sangmin Lee, Patrick T. J. Hwang, Kee-Joon Lee, Jae-Kook Cha* , Sung-Hwan Choi* , and Jinkee Hong*

General comments

- Symbiotically Integrating Occlusive Membrane for Guided Regeneration of Inflammatory-Challenged Complex Tissue Defect, by Choi et al. has been improved with the addition of experiments and more detail. The authors look at membrane effects on healing and associated oral microbiome. I appreciate the authors' efforts in this resubmission. I just have a few minor requests remaining on this work that in my view is near ready for acceptance.
- It is good that they were able to show that in the canine model there is also a reduction in richness of amplicon sequence read variants as one might expect. It is good that they use ANCOM-BC a program for looking at differential sequence abundance that as that corrects well technical differences in samples.

- That authors have demonstrated large difference in microbiome associated with CM, ZM and SIOM. They have improved their case that SIOM retards *P. gingivalis* and other pathogens *S. aureus*.

Authors' response

The authors sincerely appreciate your tremendous contribution. Also, we thank you for acknowledging our efforts. As mentioned in the first response letter, while contemplating your comments, we were able to learn a more accurate methodology for studying the microbiome information and constructing firm logical discussion. Recognizing the merits of your comments, we thoroughly addressed the emerging requests and grew once more. The following are the specific responses to your comments. The authors heartily look forward to the favorable responses.

Comment #1

“For the amplicon sequencing, the Illumina Miseq Sequencing System from Illumina, USA was utilized. To ensure data integrity, a quality check on the raw reads, filtering out low-quality reads with a score of less than 25 was performed. Paired-end sequence data was merged and the primers were trimmed. The unique 16S rRNA reads were isolated at similarity threshold of 97%, and allocated to taxonomy based on the EzBioCloud 16S rRNA database”

In the supplement there needs to be at least a reference to the method used to merge paired sequence and the identification of similar sequence reads(DADA2?) unless it is elsewhere in the paper and I missed it.

Authors’ response to Comment #1

We sincerely thank you for the helpful suggestion and apologize for the oversight. To improve the accuracy of the methodology description, the manuscript has been revised, reflecting the reviewer’s kind feedback, as shown below.

Changes in the manuscript reflecting Comment #1

(SI, Page 6)

For the amplicon sequencing, the Illumina Miseq Sequencing System from Illumina, USA was utilized. To ensure data integrity, a quality check on the raw reads, filtering out low-quality reads with a score of less than 25 was performed. The Divisive Amplicon Denoising Algorithm 2 (DADA2 version 1.26) was applied within R Studio (R version 4.2), to conduct the assembly of paired-end reads and the subsequent

assignment of these reads to amplicon sequence variants. The unique 16S rRNA reads were isolated at similarity threshold of 97%, and allocated to taxonomy based on the EzBioCloud 16S rRNA database.

Comment #2

I do not understand supplemental table 4. If FDR is >0.1 for a comparison that is usually not considered significantly differentially abundant. Like in the case of Actinobacteria. If another system is used to define statistical significance it needs to be even more clearly expressed.

Authors' response to Comment #2

The authors fully concur with the reviewer's difficulty in interpreting the False Discovery Rate (FDR). As depicted in the revised Supplementary Table 4, we have presented the ANCOM-BC comparison results for differential abundance. This table offers a comprehensive overview of the outcomes of the statistical test, aiding readers in gaining a thorough understanding of the results, in conjunction with Supplementary Fig. 20.

In line with the valuable suggestion from the reviewer, we have incorporated an FDR threshold of 0.05 to limit the proportion of false positives in significant results to 5%. To enhance the table's clarity, we have updated it to explicitly indicate statistically significant differences in abundance by marking them with asterisks (*).

Changes in the manuscript reflecting Comment #2

(SI, Page 34)

Supplementary Table 4. FDR (false discovery rate) bias-corrected analysis of differentially abundant taxa with analysis of compositions of microbiomes with bias correction (ANCOM-BC)

Phyla	Beta-coefficient (W)		Standard error		Differentially abundant		Relative abundance		FDR value (q)	
	CM	SIOM	CM	SIOM	CM	SIOM	CM	SIOM	CM	SIOM
Tenericutes	-2.08695	-0.86879184	0.530866	0.63809377	FALSE	FALSE	3.33	1.33	5.74E-02	4.49E-01
Spirochaetes	-2.090111	-3.75868513	1.02842	0.76510948	FALSE	TRUE	197.33	44.00	5.74E-02	5.86E-04*
Proteobacteria	1.160967	0.28546156	0.695246	0.47726797	FALSE	FALSE	1374.00	852.00	3.44E-01	7.75E-01
Fusobacteria	0.220584	3.07034844	0.834404	0.32180151	FALSE	TRUE	9678.67	10601.00	8.25E-01	4.28E-03*
Bacteroidetes	0.500062	1.2329256	0.374642	0.31016438	FALSE	FALSE	10836.67	10717.33	6.81E-01	3.05E-01
Acidobacteria	3.215367	3.33564011	0.448713	0.327173	TRUE	TRUE	108.33	132.67	3.04E-03*	1.99E-02
Verrucomicrobia	4.633183	3.707471	0.21341	0.27674468	TRUE	TRUE	0.00	0.00	1.68E-05*	5.86E-04*
Synergistetes	-9.247514	-9.62984912	0	0	TRUE	TRUE	0.00	0.33	0.00E+00*	0.00E+00*
Planctomycetes	4.692319	4.20788622	0.231824	0.26213618	TRUE	TRUE	0.33	0.33	1.68E-05*	1.20E-04*
Other	-2.463879	-1.71939438	0.658528	0.8269246	TRUE	FALSE	14.33	176.33	2.75E-02*	1.33E-01
Firmicutes	3.963607	5.21980338	0.283852	0.22013747	TRUE	TRUE	11653.00	10296.00	2.58E-04*	1.25E-06*
Cyanobacteria	3.278702	2.91790054	0	0	TRUE	TRUE	1.67	2.67	0.00E+00*	0.00E+00*
Chloroflexi	-0.47846	-0.47881451	0	0	TRUE	TRUE	0.00	0.00	0.00E+00*	0.00E+00*
Actinobacteria	-1.041663	-1.02959918	0.588714	0.56279316	FALSE	FALSE	108.33	132.67	3.79E-01	3.86E-01

#ANCOMBC2 identifies taxa that exhibit significant differences in abundance between groups. The beta coefficient (*W*), is a measure of the effect size representing the change in the relative abundance of a taxon between the groups being compared. It

provides an estimate of the magnitude and direction of change in abundance associated with the group. The magnitude and direction of W indicate the degree of differential abundance. Larger absolute values of W suggest stronger differential abundance, while values closer to zero indicate less pronounced differences in abundance between groups. FDR (q) values present the adjusted p values by Benjamini & Hochberg correction of false discovery rate. The significant threshold was set at $q < 0.05$. The asterisk symbol in q values represents the truly differentially abundant Phyla.

Comment #3

“However, the oral microbe-related material technologies are still bound^{5, 16} to bacterial-free functionalities, for instance, antifouling 49, 74 and bactericidal effects”

This is just a suggestion but may want to change that to: However, the oral microbe-related material technologies still are focused on total prevention of bacteria accumulation, for instance, antifouling 49, 74 and bactericidal effects 75,76,77.

The above may or may not be clearer.

Authors' response to Comment #3

The authors gratefully appreciate your opinion. We entirely accept your recommendation to improve comprehension. We ultimately reflected on your comment as shown below.

Changes in the manuscript reflecting Comment #3

(Page 27)

However, oral microbe-related material technologies are still focused on total prevention of bacteria accumulation, for instance, antifouling and bactericidal effects.

REVIEWERS' COMMENTS

Reviewer #2 (Remarks to the Author):

Symbiotically Integrating Occlusive Membrane for Guided Regeneration of Inflammatory-Challenged Complex Tissue Defects, by Choi et al. is an interesting, well constructed paper that is now ready for publication.